# Chromatin structure-dependent histone incorporation revealed by a genome-wide deposition assay

Hiroaki Tachiwana[1]*, Mariko Dacher[2], Kazumitsu Maehara[3], Akihito Harada[3], Yosuke Seto[4], Ryohei Katayama[4], Yasuyuki Ohkawa[3], Hiroshi Kimura[5], Hitoshi Kurumizaka[2], Noriko Saitoh[1]*

[1]Division of Cancer Biology, The Cancer Institute of Japanese Foundation for Cancer Research, Tokyo, Japan; [2]Laboratory of Chromatin Structure and Function, Institute for Quantitative Biosciences, The University of Tokyo, Tokyo, Japan; [3]Division of Transcriptomics, Medical Institute of Bioregulation, Kyushu University, Fukuoka, Japan; [4]Division of Experimental Chemotherapy, Cancer Chemotherapy Center, Japanese Foundation for Cancer Research, Tokyo, Japan; [5]Cell Biology Center, Institute of Innovative Research, Tokyo Institute of Technology, Yokohama, Japan

**Abstract** In eukaryotes, histone variant distribution within the genome is the key epigenetic feature. To understand how each histone variant is targeted to the genome, we developed a new method, the RhIP (*R*econstituted *h*istone complex *I*ncorporation into chromatin of *P*ermeabilized cell) assay, in which epitope-tagged histone complexes are introduced into permeabilized cells and incorporated into their chromatin. Using this method, we found that H3.1 and H3.3 were incorporated into chromatin in replication-dependent and -independent manners, respectively. We further found that the incorporation of histones H2A and H2A.Z mainly occurred at less condensed chromatin (open), suggesting that condensed chromatin (closed) is a barrier for histone incorporation. To overcome this barrier, H2A, but not H2A.Z, uses a replication-coupled deposition mechanism. Our study revealed that the combination of chromatin structure and DNA replication dictates the differential histone deposition to maintain the epigenetic chromatin states.

*For correspondence:
hiroaki.tachiwana@jfcr.or.jp (HT);
noriko.saito@jfcr.or.jp (NS)

## Introduction

Eukaryotic genomic DNA is packaged into chromatin, in which the basic structural unit is the nucleosome. The nucleosome is composed of around 150 base pairs of DNA and a histone octamer consisting of two copies of each core histone, H2A, H2B, H3, and H4 (*Luger et al., 1997*). Chromatin is not only the storage form of genomic DNA but also the regulator of DNA-templated processes, such as transcription, replication, repair, and chromosome segregation. To enable these processes, chromatin forms various structures that can be reversibly altered. Historically, cytological studies first identified euchromatin and heterochromatin, which are transcriptionally active and inactive regions, respectively (*Passarge, 1979*). Subsequent studies revealed that euchromatin coincides with nuclease hypersensitivity, indicating that it forms a more accessible structure (open chromatin) than heterochromatin (closed chromatin) (*Garel and Axel, 1976*; *Spiker et al., 1983*; *Tsompana and Buck, 2014*; *Weintraub and Groudine, 1976*). Although these chromatin structures are involved in the regulation of DNA template-mediated processes, little is known about how the open and closed chromatin configurations are formed and maintained.

Among the chromatin associated proteins, histones have a significant impact on the chromatin structure. In humans, the canonical histones, H2A, H2B, and H3.1, have non-allelic variants with

distinct expression and/or localization patterns (*Buschbeck and Hake, 2017*; *Maehara et al., 2015*). Canonical H3.1 and H2A are expressed in S phase and show genome-wide localizations (*Buschbeck and Hake, 2017*; *Wu et al., 1982*). In contrast, the histone variants H3.3 and H2A.Z are expressed throughout the cell cycle and concentrated at promoters in both open chromatin and pericentric heterochromatin (*Ahmad and Henikoff, 2002*; *Boyarchuk et al., 2014*; *Drané et al., 2010*; *Goldberg et al., 2010*; *Greaves et al., 2007*; *Jin and Felsenfeld, 2007*; *Raisner et al., 2005*; *Rangasamy et al., 2003*; *Sarcinella et al., 2007*; *Wu et al., 1982*). In addition, H3.3 localizes at telomeres (*Goldberg et al., 2010*). Another H2A variant, MacroH2A, is found at transcriptionally suppressed chromatin, such as inactive X chromosomes (*Costanzi et al., 2000*). Along with the canonical H2A, the localization of H2A.X, which functions in DNA repair, is genome-wide (*Yukawa et al., 2014*). Thus, in spite of the high-sequence homology between the canonical and variant forms, each histone has specific functions. A previous study identified the six essential residues responsible for the H2A.Z-specific functions, which are located in the αC helix called the M6 region (*Clarkson et al., 1999*). The H2A.Z swap mutant, in which M6 was replaced with the equivalent residues in H2A, failed to rescue the embryonic lethality of the H2A.Z null mutant in *Drosophila melanogaster*. Thus, the importance of M6 in the organism was evident; however, its molecular mechanism remained elusive.

Histone deposition and exchange have been studied for more than 30 years (*Clément et al., 2018*; *Jackson, 1990*; *Jansen et al., 2007*; *Kimura and Cook, 2001*; *Louters and Chalkley, 1985*; *Ray-Gallet et al., 2011*; *Tachiwana et al., 2010*). An in vivo histone deposition study must distinguish the pre-incorporated histones (parental histones) from the newly incorporated histones (new histones). To enable this, initial studies used radio-labeled histones and revealed that histone deposition occurred by both replication-coupled and replication-independent mechanisms (*Jackson, 1990*; *Louters and Chalkley, 1985*). In principle, the replication-independent histone deposition would accompany histone eviction from chromatin, in a process described as histone exchange or turnover. Several key proteins involved in these mechanisms have been identified. Anti-silencing function 1 (ASF1), together with chromatin assembly factor 1 (CAF-1), plays indispensable roles in replication-coupled H3.1 deposition (*Smith and Stillman, 1989*; *Tagami et al., 2004*; *Tyler et al., 1999*; *Tyler et al., 1996*; *Tyler et al., 2001*). CAF-1 is a heterotrimeric complex composed of p150, p60, and p48, and directly interacts with proliferating cell nuclear antigen (PCNA), leading to the assembly of H3.1 at replicating chromatin (*Moggs et al., 2000*; *Shibahara and Stillman, 1999*). The histone regulator A (HIRA) or death-associated protein (DAXX) promotes the accumulation of H3.3 at transcription sites and regulatory elements in a replication-independent manner (*Drané et al., 2010*; *Goldberg et al., 2010*; *Ray-Gallet et al., 2011*; *Ray-Gallet et al., 2002*; *Tagami et al., 2004*).

H2A- and H2A.Z-specific chaperones have also been identified (*Obri et al., 2014*). Facilitates chromatin transcription (FACT), a histone chaperone, may be involved in replication-coupled H2A deposition (*Orphanides et al., 1998*; *Ransom et al., 2010*; *Wittmeyer and Formosa, 1997*). Recent studies revealed that FACT binds to the H3-H4 complex, indicating that FACT functions in H3-H4 depositions as well (*Liu et al., 2020*; *Tsunaka et al., 2016*; *Wang et al., 2018*). ANP32E functions in H2A.Z eviction from chromatin, and the SNF2-related CBP activator protein (SRCAP) chromatin-remodeling subunit YL1 promotes H2A.Z deposition at gene promoters (*Latrick et al., 2016*; *Liang et al., 2016*; *Mao et al., 2014*; *Obri et al., 2014*; *Wong et al., 2007*). The histone acetyl transferase (HAT) complex, NuA4/TIP60, also functions in H2A.Z deposition by interacting with pre-deposited H2A.Z (*Gévry et al., 2009*; *Giaimo et al., 2019*; *Shia et al., 2006*).

The precise distribution of histones is crucial for chromatin organization and its epigenetic states. The ChIP-seq analysis method is powerful and useful to visualize steady state histone localizations; however, it does not enable the analysis of the incorporation of each histone into open/closed chromatin or both (genome-wide). To analyze histone incorporations, several approaches using fluorescence imaging, mass spectrometry, inducible tagged proteins, and labeling of newly synthesized proteins have been developed (*Deal and Henikoff, 2010*). Imaging-based methods, such as fluorescence recovery after photobleaching (FRAP) and SNAP/CLIP-tag technology, were developed to analyze the histone dynamics in living cells. FRAP is suitable for investigating the histone mobility in living cells (*Kimura and Cook, 2001*; *Tachiwana et al., 2010*). The SNAP/CLIP-technology is effective for analyzing the histone deposition, as it can distinguish parental histones from new histones in cells with cell-permeable fluorophores that covalently bind to the tag (*Clément et al., 2018*; *Jansen et al., 2007*; *Ray-Gallet et al., 2011*). Although these are powerful methods, they have

resolution limitations since they are based on imaging. The SNAP/CLIP-technology led to the establishment of the time-ChIP method, in which a biotin-labeled SNAP-histone is captured (*Deaton et al., 2016*). The time-ChIP method was developed to measure the stability of parental histones, rather than the histone incorporation, as it has a time lag during the synthesis and labeling of histones, which limits the time resolution (*Deaton et al., 2016*; *Siwek et al., 2018*). The tetracycline (Tet) inducible expression system, which induces the expression of epitope-tagged histones, has been applied for analyses of nucleosome turnover/histone exchange. However, this system also has a time resolution limitation (*Ha et al., 2014*; *Yildirim et al., 2014*). To uncover the correlation between histone incorporation and open/closed chromatin structures, a new analysis method for the histone incorporation at the DNA sequence level is desired.

Permeabilized cells are useful to dissect the molecular pathways in nuclear events (*Adam et al., 1990*; *Okuno et al., 1993*). In assays with such cells, the cellular membranes are permeabilized by a nonionic detergent treatment. In the permeabilized cells, the chromatin and nuclear structures remain intact and react with exogenously added proteins (*Kimura et al., 2006*; *Maison et al., 2002*; *Misteli and Spector, 1996*; *Saitoh et al., 2006*). A previous study showed that an exogenously added GFP-tagged histone, prepared from cultured human cells, was incorporated into the chromatin of permeabilized cells in the presence of a cellular extract (*Kimura et al., 2006*). Moreover, permeabilized cells are suitable for monitoring replication timing, by labeling the nascent DNA with exogenously introduced nucleotides (*Kimura et al., 2006*; *Misteli and Spector, 1996*).

In the present study, we developed a new method, in which a reconstituted histone complex, instead of a fluorescent protein-tagged histone, was added to permeabilized cells. We named this the RhIP assay (*R*econstituted *h*istone complex *I*ncorporation into chromatin of *P*ermeabilized cell). Since the histone complexes are reconstituted in vitro using epitope-tagged recombinant histones, RhIP with sequencing allows the analysis of incorporations at the DNA sequence level, without the need for specific antibodies. We found that the chromatin structure regulates the histone incorporations, which may be necessary for maintaining the epigenetic state of chromatin.

## Results

### RhIP assay reproduces in vivo histone deposition

To understand how histones are incorporated into chromatin in cells, we developed the RhIP assay, in which an in vitro reconstituted histone complex, nucleotides, and a cellular extract are added to permeabilized cells (*Figure 1A*). We first confirmed that the RhIP assay can recapitulate the specific histone incorporations observed in cells. The H3.1-H4 incorporation into chromatin is coupled with replication, while the H3.3-H4 incorporation occurs throughout the cell cycle (*Ahmad and Henikoff, 2002*). We reconstituted H3-H4 complexes in vitro, using recombinant H3.1, H3.3, and H4 (*Figure 1B*). The recombinant H3.1 and H3.3 were fused to HA and FLAG tags at their C-termini, respectively. The permeabilized cells were then prepared by treating HeLa cells with a nonionic detergent, Triton X-100, and the reconstituted H3-H4 complexes were mixed with the cellular extract and nucleotides. Cy5-dUTP was also added, in order to monitor DNA replication. After the reaction, the exogenously added H3.1 and H3.3 were detected with antibodies against the HA and FLAG tags, respectively (*Figure 1C–E*). As a result, the H3.1 was detected in the Cy5 positive cells (S phase cells), while the H3.3 was detected irrespective of the Cy5 signal. As the co-incubated H3.1–3HA-H4 and H3.3-3FLAG-H4 complexes showed different staining patterns, they were incorporated into the chromatin by specific mechanisms, rather than non-specifically, in the RhIP assay.

To improve the resolution of the analysis, we performed the RhIP assay followed by chromatin immunoprecipitation (RhIP-ChIP) (*Figure 1F*). The reconstituted H3.1–3HA-H4 or H3.3–3HA-H4 complex was added to permeabilized cells along with the cellular extract and nucleotides, including Cy5-dUTP to label the nascent DNA (*Figure 1F and G*). After the reaction, the chromatin was partially digested by micrococcal nuclease (MNase), and the nucleosomes containing H3.1–3HA or H3.3–3HA were immunoprecipitated with an antibody against the HA tag. The precipitated DNA was then extracted and analyzed by gel electrophoresis. As shown in *Figure 1H*, the amounts of precipitated DNA are nearly the same between the H3.1 and H3.3 samples, as judged from the SYBR Gold staining (*Figure 1H*, upper); however, the amount of nascent DNA labeled with Cy5 is much greater in the H3.1 sample than in the H3.3 sample (*Figure 1H*, lower). This result indicates that H3.1 is

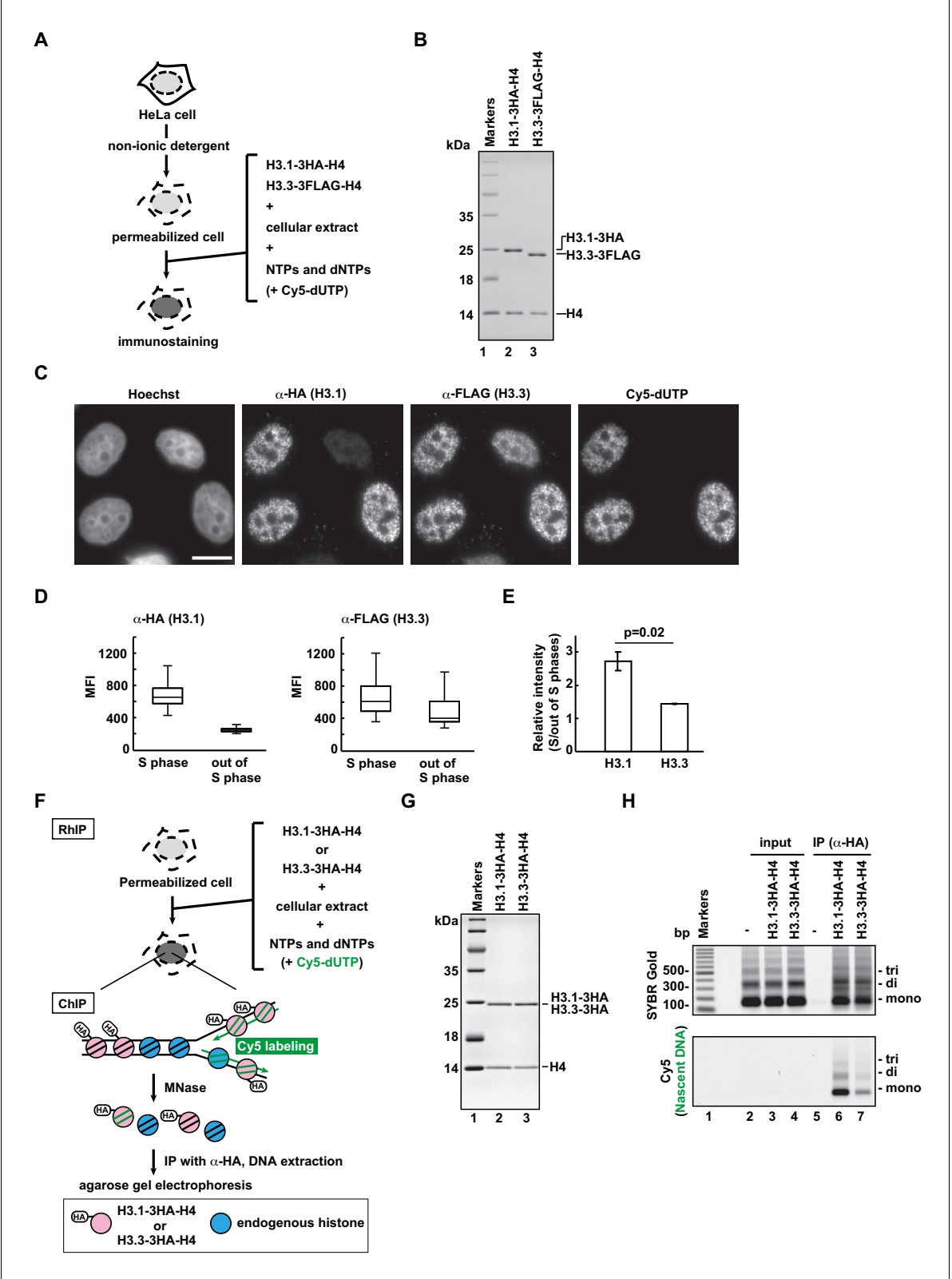

**Figure 1.** RhIP (*Reconstituted histone complex Incorporation into chromatin of Permeabilized cells*) assay recapitulates the replication-coupled H3.1-H4 and -dependent H3.3-H4 depositions. (**A**) Schematic representation of the RhIP assay, using reconstituted H3.1-H4 and H3.3-H4 complexes. Permeabilized cells were prepared from HeLa cells treated with non-ionic detergent, to perforate the cellular membranes. The in vitro reconstituted H3-H4 complexes were then added to the cells with the cellular extract and nucleotides. Cy5-dUTP was added to label the nascent DNA, so replication

*Figure 1 continued on next page*

*Figure 1 continued*

could be monitored. (**B**) Reconstituted H3.1-H4 and H3.3-H4 complexes were analyzed by SDS-16% PAGE with Coomassie Brilliant Blue staining. The 3HA and 3FLAG tags were fused to the C-termini of H3.1 and H3.3, respectively. Lane one indicates the molecular mass markers, and lanes 2 and 3 indicate the H3.1-H4 and H3.3-H4 complexes, respectively. (**C**) RhIP-immunostaining of H3.1 and H3.3. Exogenously added H3-H4 complexes were stained with an anti-HA or -FLAG antibody. Cells in S phase were monitored with Cy5-dUTP, which was incorporated into the nascent DNA. Bar indicates 10 µm. (**D**) Quantification of C. The mean fluorescence intensities (MFI) of H3.1–3HA (left) and H3.3-3FLAG (right) were measured. Nuclei were divided into S phase (Cy5 positive) and out of S phase (Cy5 negative) (n > 50, triplicate). (**E**) Relative intensity of H3.1 or H3.3 signal in S phase against signal out of S phase. Experiments were repeated three times and averaged data with standard deviations are shown. The two-tailed Student's t-test was used for the statistical comparisons. (**F**) Schematic representation of the RhIP-ChIP assay, using the reconstituted H3.1-H4 and H3.3-H4 complexes. The reconstituted H3.1-H4 or H3.3-H4 complex was added to permeabilized cells with the cellular extract and nucleotides. Cy5-dUTP was added to label the nascent DNA. The chromatin was partially digested with micrococcal nuclease (MNase). Chromatin immunoprecipitation was performed with anti-HA magnetic beads. The precipitated DNA was extracted and analyzed by agarose gel electrophoresis. (**G**) Reconstituted H3.1-H4 and H3.3-H4 complexes were analyzed by SDS-16% PAGE with Coomassie Brilliant Blue staining. A 3HA tag was fused to the C-termini of H3.1 and H3.3. Lane one indicates the molecular mass markers, and lanes 2 and 3 indicate the H3.1-H4 and H3.3-H4 complexes, respectively. (**H**) The immunoprecipitated DNA was analyzed by 2% agarose electrophoresis. Upper and lower images were obtained from the same gel. The DNA was visualized with SYBR Gold (upper), and the nascent DNA was visualized by detecting the Cy5 signals (lower). Lane 1 indicates the 100 bp DNA ladder. Lanes 2–4 and 5–7 indicate input samples and immunoprecipitated samples, respectively. Each set includes the experiments with no reconstituted histone complex (negative control, lanes 2 and 5), with H3.1–3HA-H4 (lanes 3 and 6), and with H3.3–3HA-H4 (lanes 4 and 7).

The online version of this article includes the following figure supplement(s) for figure 1:

**Figure supplement 1.** H3.3 is enriched in active genes in the RhIP assay.

**Figure supplement 2.** CAF-1 and HIRA in permeabilized cells are essential for H3.1 and H3.3 incorporations in the RhIP assay.

**Figure supplement 3.** The exogenously added H3.1 and H3.3 are mostly incorporated into the chromatin in the RhIP assay.

**Figure supplement 4.** Replication-coupled H3.1 deposition in the RhIP assay requires the cellular extract.

incorporated into replicating chromatin more efficiently than H3.3. We further performed the RhIP-ChIP-seq of H3.3 to examine the distribution of incorporated H3.3 in the RhIP assay (*Figure 1—figure supplement 1A*). We found that exogenously added H3.3 was preferentially incorporated into transcriptionally active genes, rather than inactive genes. This distribution pattern is consistent with many previous studies (*Bachu et al., 2019*; *Mito et al., 2005*; *Pchelintsev et al., 2013*). The RhIP-ChIP-qPCR analysis also demonstrated the H3.3 enrichment at the transcriptionally active GAPDH, as compared to that at the inactive gene, LNC02199 (*Figure 1—figure supplement 1B and C*). Thus, our results imply that the reconstituted H3.1-H4 and H3.3-H4 complexes are incorporated into the chromatin of permeabilized cells with the same dynamics as observed in intact cells.

CAF-1 and HIRA are key factors for the H3.1 and H3.3 depositions in vivo, respectively (*Drané et al., 2010*; *Goldberg et al., 2010*; *Ray-Gallet et al., 2002*; *Ray-Gallet et al., 2011*; *Smith and Stillman, 1989*; *Tagami et al., 2004*; *Tyler et al., 1996*; *Tyler et al., 1999*; *Tyler et al., 2001*). We therefore examined whether their roles can be recapitulated in the RhIP assay (*Figure 1—figure supplement 2*). We first analyzed whether CAF-1 and HIRA remain in the permeabilized cells or are extracted during the permeabilization process (*Figure 1—figure supplement 2A*). Our immunoblot showed that both CAF-1 (p60) and HIRA remained in the permeabilized cells. We then performed the RhIP-immunostaining assay of H3.1 or H3.3 using CAF-1- or HIRA-knockdown cells, respectively (*Figure 1—figure supplement 2B*). For a precise evaluation, we co-cultured the control and knockdown cells on the same coverslips, and performed RhIP-immunostaining assays of H3.1 or H3.3. The knockdown cells were judged by immunostainings for CAF-1 (p60) (cells with arrowheads in *Figure 1—figure supplement 2C*) and HIRA (cells with arrowheads in *Figure 1—figure supplement 2D*). For the detection of H3.1, cells were synchronized to S phase by adding thymidine 18 hr before performing the RhIP assay. As a result, the H3.1 incorporation was detected in CAF-1-positive S phase cells (arrows in *Figure 1—figure supplement 2C*). Similarly, the H3.3 was detected in HIRA positive cells (arrows in *Figure 1—figure supplement 2D*). We also investigated whether H3.1 and H3.3 were incorporated into the chromatin or merely bound to histone chaperones, CAF-1 and HIRA, respectively, which are present in the chromatin of the permeabilized cells. After performing the RhIP assay, we washed the cells with PBST containing 300 mM NaCl prior to the fixation and immunostaining processes (*Figure 1—figure supplement 3A*). We found that that the exogenously added H3.1 or H3.3 was still detected after washing the cells with PBST containing 300 mM NaCl, even though CAF-1 and HIRA were no longer detected (*Figure 1—figure supplement 3B–E*). Therefore, the exogenously added H3.1 and H3.3 were mostly incorporated into the chromatin of the

permeabilized cells in the RhIP assay. These data demonstrated that the H3.1 and H3.3 incorporations into chromatin in cells were correctly recapitulated in the RhIP assay.

We further examined whether the cellular extract is essential or replaceable by the histone chaperones, NAP1 or ASF1 (*Figure 1—figure supplement 4A*). NAP1 and ASF1 bind to the H2A-H2B and H3-H4 complexes in vivo, respectively, and both promote nucleosome formation in vitro (*Ishimi et al., 1984*; *Munakata et al., 2000*; *Tachiwana et al., 2008*; *Tyler et al., 1999*). Human NAP1 and ASF1 were purified as recombinant proteins (*Figure 1—figure supplement 3B*). The H3.1-H4 complex was then added to the permeabilized cells in the absence of the cellular extract or in the presence of NAP1 or ASF1, and the incorporation was analyzed by immunostaining (*Figure 1—figure supplement 4C*). Without the cellular extract or the histone chaperone, the exogenously added H3.1 was not detected in the permeabilized cells, indicating that no H3.1 incorporation had occurred. NAP1 promoted the promiscuous incorporation irrespective of DNA replication, and ASF1 facilitated H3.1 accumulation in the nucleoli. These data indicated that the functional deposition of the exogenously added histone complex requires the cellular extract, which may contain essential components. Together with the fact that the efficiency of RhIP-ChIP of exogenously added H3.3 is almost the same as that of endogenous H3.3 ChIP (*Figure 1—figure supplement 1D and E*), we conclude that the RhIP assay reproduces cellular histone deposition and is suitable for analyzing histone incorporation in vitro.

## H2A.Z incorporation into chromatin differs from that of H2A and H2A. X

Among the H2A family members, canonical H2A and the H2A.X variant show even and broad genome-wide distributions, but H2A.Z specifically localizes in open chromatin, including promoters and enhancers (*Buschbeck and Hake, 2017*; *Raisner et al., 2005*). To test whether this difference reflects their deposition manners, we performed the RhIP assay (*Figure 2A*). The H2A-H2B and H2A. Z-H2B complexes were reconstituted in vitro using recombinant proteins (*Figure 2B*) and added to permeabilized cells, which were then immunostained (*Figure 2C*). The H2A and H2A.Z signals were both observed in the Cy5-negative and -positive permeabilized cells, indicating that their incorporations occur irrespective of DNA replication. We also found that H2A forms foci in the S phase nuclei. We then merged the images of the H2A and Cy5 signals. The replication foci change as cells progress through S phase (*Leonhardt et al., 2000*). In early S phase, the replication foci are present throughout the nucleoplasm, except for the nucleoli. The foci then accumulate at the nuclear periphery and around the nucleoli. In late S phase, the foci increase in size but decrease in number. We found that the H2A signals overlapped well with the replication foci throughout replication (*Figure 2C and D*). The H2A.Z signals overlapped with the early replication foci to some extent, but they were clearly eliminated from the late replication foci (*Figure 2C and D*). In contrast to the difference between H2A and H2A.Z, the H2A and H2A.X signals overlapped well with each other, suggesting that their incorporation mechanism is the same (*Figure 2—figure supplement 1*). These results indicate that H2A, H2A.X, and H2A.Z can be incorporated into chromatin in a replication-independent manner; however, during S phase, H2A and H2A.X are preferentially incorporated into the chromatin of ongoing replication sites, in contrast to H2A.Z.

We further analyzed the incorporation of H2A and H2A.Z into replicating chromatin by a RhIP-ChIP assay, as in *Figure 1F* (*Figure 2E and F*). The amounts of precipitated DNA are nearly the same between the H2A and H2A.Z precipitants, as judged from the SYBR Gold staining (*Figure 2F*, upper); however, the amount of nascent DNA labeled with Cy5 is much greater in the H2A precipitant than in the H2A.Z sample (*Figure 2F*, lower). This result indicates that H2A is incorporated into replicating chromatin more efficiently than H2A.Z.

The replication timing in S phase strongly correlates with the chromatin configurations (*Rivera-Mulia and Gilbert, 2016*). In general, early and late replicating chromatin regions correspond to open and closed chromatin, respectively. We investigated whether the efficiencies of H2A and H2A. Z incorporation into replicating chromatin change, according to the replication timing (*Figure 2—figure supplement 2*). For this analysis, the cells were synchronized in early S phase by a double thymidine block, and then early and late S phase cells were collected at 0 and 5 hr post thymidine-release, respectively. The synchronized cells showed the typical early and late replication foci representing nascent DNA labeled with Cy5-dUTP (*Figure 2—figure supplement 2A*). Using these cells, we performed RhIP-ChIP assays of H2A and H2A.Z. The results revealed that the incorporation

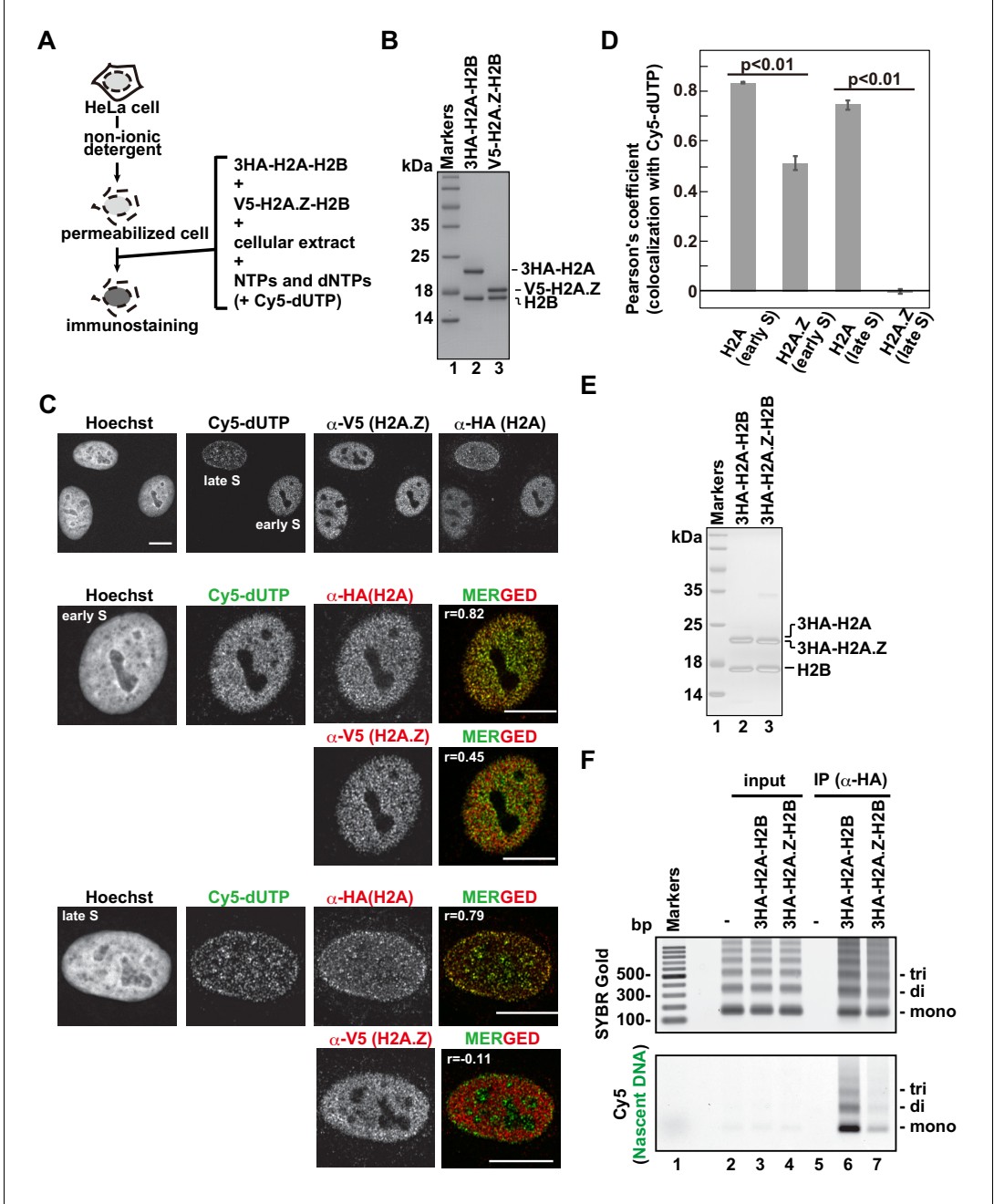

**Figure 2.** The H2A.Z and H2A deposition patterns are different in the RhIP assay. (**A**) Schematic representation of the RhIP assay, using the reconstituted H2A-H2B and H2A.Z-H2B complexes. (**B**) The reconstituted H2A-H2B and H2A.Z-H2B complexes were analyzed by SDS-16% PAGE with Coomassie Brilliant Blue staining. The 3HA and V5 tags were fused to the N-termini of H2A and H2A.Z, respectively. Lane one indicates the molecular mass markers, and lanes 2 and 3 indicate the H2A-H2B and H2A.Z-H2B complexes, respectively. (**C**) RhIP-immunostaining of H2A and H2A.Z. Top: Exogenously added H2A-H2B or H2A.Z-H2B complexes were stained with either an anti-HA or -V5 antibody. Cells in S phase were monitored with Cy5-dUTP. Middle and Bottom: merged images of Cy5-dUTP (green) and H2A or H2A.Z (red) in early S (Middle) and late S (Bottom) phase. Bar indicates 10 μm, and r indicates the Pearson's correlation coefficient. (**D**) Colocalization analyses of Cy5-dUTP and H2A.Z or H2A (n > 35 cells). Experiments were repeated three times and averaged data are shown. The two-tailed Student's t-test was used for the statistical comparisons. (**E**) Reconstituted H2A-H2B and H2A.Z-H2B complexes were analyzed by SDS-16% PAGE with Coomassie Brilliant Blue staining. A 3HA tag was fused to the N-termini of H2A and HA.Z. Lane 1 indicates the molecular mass markers, and lanes 2 and 3 indicate the H2A-H2B and H2A.Z-H2B complexes, respectively. (**F**) The RhIP-ChIP assay was performed using H2A and H2A.Z, as described in *Figure 1F*. The immunoprecipitated DNA was analyzed by 2% agarose electrophoresis. Upper and lower images were obtained from the same gel. The DNA was visualized with SYBR Gold (upper), and the nascent DNA was visualized by detecting the Cy5 signals (lower). Lane 1 indicates the 100 bp DNA ladder. Lanes 2–4 and 5–7 indicate input samples and immunoprecipitated samples,

*Figure 2 continued on next page*

*Figure 2 continued*

respectively. Each set has the experiments with no reconstituted histone complex (negative control, lanes 2 and 5), with 3HA-H2A-H2B (lanes 3 and 6), and with 3HA-H2A.Z-H2B (lanes 4 and 7).

The online version of this article includes the following figure supplement(s) for figure 2:

**Figure supplement 1.** H2A.X shows the same deposition patterns as H2A in the RhIP assay.

**Figure supplement 2.** Replication-coupled H2A.Z deposition does not change during S phase progression.

efficiencies of both H2A and H2A.Z into replicating chromatin did not change, irrespective of the replication timing in S phase. This implies that the efficiencies of replication-coupled histone deposition are not different between open (early S-replicating) and closed (late S-replicating) chromatin (*Figure 2—figure supplement 2B*).

## Open and closed chromatin structures regulate histone deposition

The RhIP-ChIP assay showed that the H2A.Z deposition on nascent DNAs was constant in the early and late S phases (*Figure 2F* and *Figure 2—figure supplement 2*). In contrast, the RhIP-immunostaining revealed that more signals of H2A.Z incorporation were overlapped at the early replicating foci than the late replicating foci (*Figure 2C and D*). This discrepancy may arise from the lower resolution of the immunostaining imaging. Some of the overlapping signals of H2A.Z and Cy5 in early S phase might represent the replication-independent H2A.Z deposition that occurred close to, but not exactly at, the replication sites in open chromatin. To determine whether the efficiency of the histone deposition depends on the open/closed chromatin configuration, we performed RhIP-ChIP-seq and analyzed the H2A and H2A.Z incorporations in each type of chromatin (*Figure 3*). First, we investigated the replication-independent histone deposition using asynchronous permeabilized cells, in which the majority of the cells are out of S phase. We found that the RhIP-ChIP-seq profiles of H2A and H2A.Z showed a strong correlation (0.62 Pearson correlation) and specific peaks at megabase resolution, which were not observed in the input samples (*Figure 3A*). We noticed that these patterns are similar to the DNaseI-seq results, which mapped open chromatin regions (*Tsompana and Buck, 2014*), suggesting that H2A and H2A.Z are predominantly incorporated into open chromatin regions. We then analyzed the efficiency of histone incorporations into open and closed chromatin using the chromHMM data (Core 15-state model), which classified the open and closed chromatin regions (*Ernst and Kellis, 2012*; *Roadmap Epigenomics Consortium et al., 2015*; *Figure 3B* and *Figure 3—figure supplement 1A*). The ChIP/input ratios of each chromatin region revealed that H2A and H2A.Z were efficiently incorporated into transcriptionally active open chromatin, while their incorporations into closed chromatin were inefficient. These results indicate that histone incorporation occurs mainly in the open chromatin regions in a replication-independent manner, and closed chromatin suppresses histone incorporation.

We then examined whether the efficiency of the histone incorporation into closed chromatin changes during S phase (*Figure 3C and D*). We performed RhIP-ChIP-seq with cells synchronized at late S phase. We found that the RhIP-ChIP-seq profiles between asynchronous cells and late S phase cells changed for H2A, but not for H2A.Z, indicating that H2A incorporation into chromatin is affected by replication, in contrast to H2A.Z incorporation (*Figure 3C* and *Figure 3—figure supplement 1B*). We then investigated how the incorporation efficiency of H2A into open and closed chromatin regions changes between asynchronous and late S phase cells. The results revealed that the efficiency of H2A incorporation into open chromatin decreased in late S phase (*Figure 3D*, lanes 1–7), while in contrast, the incorporation into closed chromatin increased (*Figure 3D*, lanes 8–10 and *Figure 3—figure supplement 1B*). Considering the fact that only closed chromatin is replicated in late S phase, the changes in the incorporation efficiency may be due to the changes in the replicating chromatin. Note that replication does not progress completely in the RhIP assay, and only a small fraction of closed chromatin is replicated under the conditions shown in *Figure 4E*, which is the reason why the ChIP/input ratio does not exceed one. We concluded that replication allows the incorporation of H2A, but not H2A.Z, in closed chromatin. The frequency of exchange is high in open chromatin and low in closed chromatin for H2A and H2A.Z. As a result, H2A.Z is specifically incorporated within open chromatin.

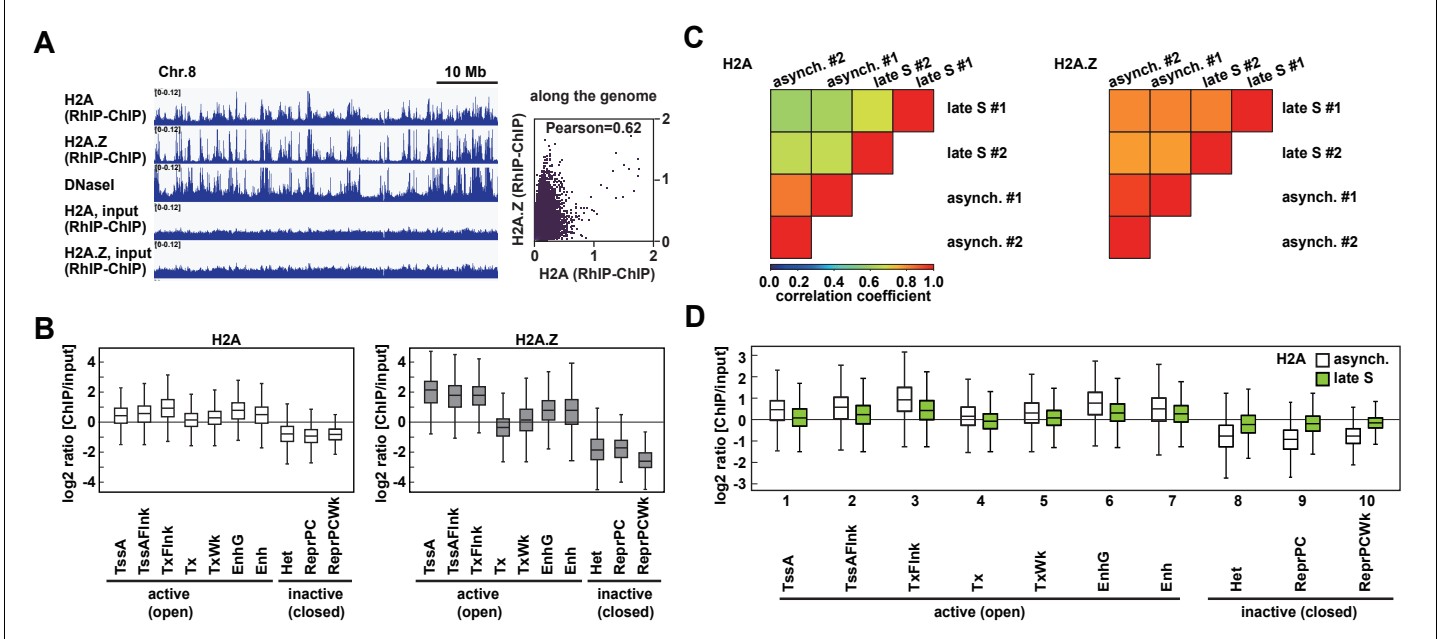

**Figure 3.** The incorporation of histones H2A and H2A. Z mainly occurs at less condensed chromatin and H2A incorporation into condensed chromatin requires a replication-coupled deposition mechanism. (**A**) RhIP-ChIP-seq and DNaseI-seq profiles using asynchronous cells were visualized with the Integrative Genomics Viewer (left). From top to bottom, profiles of H2A, H2A.Z, DNaseI-seq (GEO:GSM816643), input (H2A), and input (H2A.Z) are indicated. Scatter plot analyses of the H2A.Z and H2A RhIP-ChIP-seq along the genome (right). (**B**) Enrichment of incorporated H2A (left) or H2A.Z (right) in asynchronous cells. Each chromatin region was previously annotated by the chromHMM, as follows. TssA: active TSS, TssAFlnk: flanking active TSS, TxFlnk: transcribed state at the 5' and 3' ends of genes showing both the promoter and enhancer signatures, Tx: strong transcription, TxWk: weak transcription, EnhG: genic enhancers, Enh: enhancers, Het: heterochromatin, ReprPC: repressed PolyComb, and ReprPCWk: weak repressed PolyComb (**Ernst and Kellis, 2012**; **Roadmap Epigenomics Consortium et al., 2015**). (**C**) The Pearson's correlation coefficients between asynchronous and late S cells (two biological replicates each) were calculated from the RhIP-ChIP-seq data at 800 bp intervals. Left and right panels indicate the correlation coefficients of the H2A and H2A.Z data, respectively. (**D**) Enrichment of incorporated H2A in asynchronous (white boxes) and late S (green boxes) phase cells. Data for asynchronous cells are those shown in **Figure 4C**.

The online version of this article includes the following figure supplement(s) for figure 3:

**Figure supplement 1.** Biological replicates of RhIP-ChIP-seq analyses of H2A and H2A.Z.
**Figure supplement 2.** Quality check of RhIP-ChIP-seq analysis of H2A.Z.

## The distribution of incorporated H2A.Z in the RhIP assay is similar to the steady-state of H2A.Z localizations

As endogenous H2A.Z predominantly localizes in transcription start sites (TSS) (**Barski et al., 2007**; **Buschbeck and Hake, 2017**; **Ernst et al., 2011**; **Jin et al., 2009**; **Link et al., 2018**; **Obri et al., 2014**; **Schones et al., 2008**), we further analyzed the H2A.Z distributions in the RhIP assay at kilobase resolution (**Figure 4**). The alignment of the H2A.Z profiles from RhIP-ChIP-seq and ChIP-seq (**ENCODE Project Consortium, 2012**) at a representative chromosome position (chr10:73,079,443–73,805,171) highlighted the similarity between the two, but the H2A profiles of the RhIP-ChIP-seq did not (**Figure 4A**). Moreover, in the RhIP assay the incorporated H2A.Z predominantly accumulated at the TSS of expressed genes, as also observed in the ChIP-seq analysis of H2A.Z, but the incorporated H2A was relatively excluded from the TSS (**Figure 4B**). A correlation analysis at known H2A.Z sites, determined by a ChIP-seq analysis, showed a moderately positive, linear relationship between H2A.Z of RhIP-ChIP-seq and H2A.Z of ChIP-seq (0.31 Pearson correlation coefficient) and little to no correlation between H2A of RhIP-ChIP-seq and H2A.Z of ChIP-seq (0.20 Pearson correlation coefficient) (**Figure 4C**). In addition, a correlation analysis at the known H2A.Z sites located in gene bodies showed a fairly linear relationship between H2A.Z of RhIP-ChIP-seq and H2A.Z of ChIP-seq (0.25 Pearson correlation coefficient) (**Figure 4C**, right). The heatmap revealed that the H2A.Z of RhIP-ChIP-seq accumulates at almost all of the H2A.Z peaks found by ChIP-seq (**Figure 4D** left and

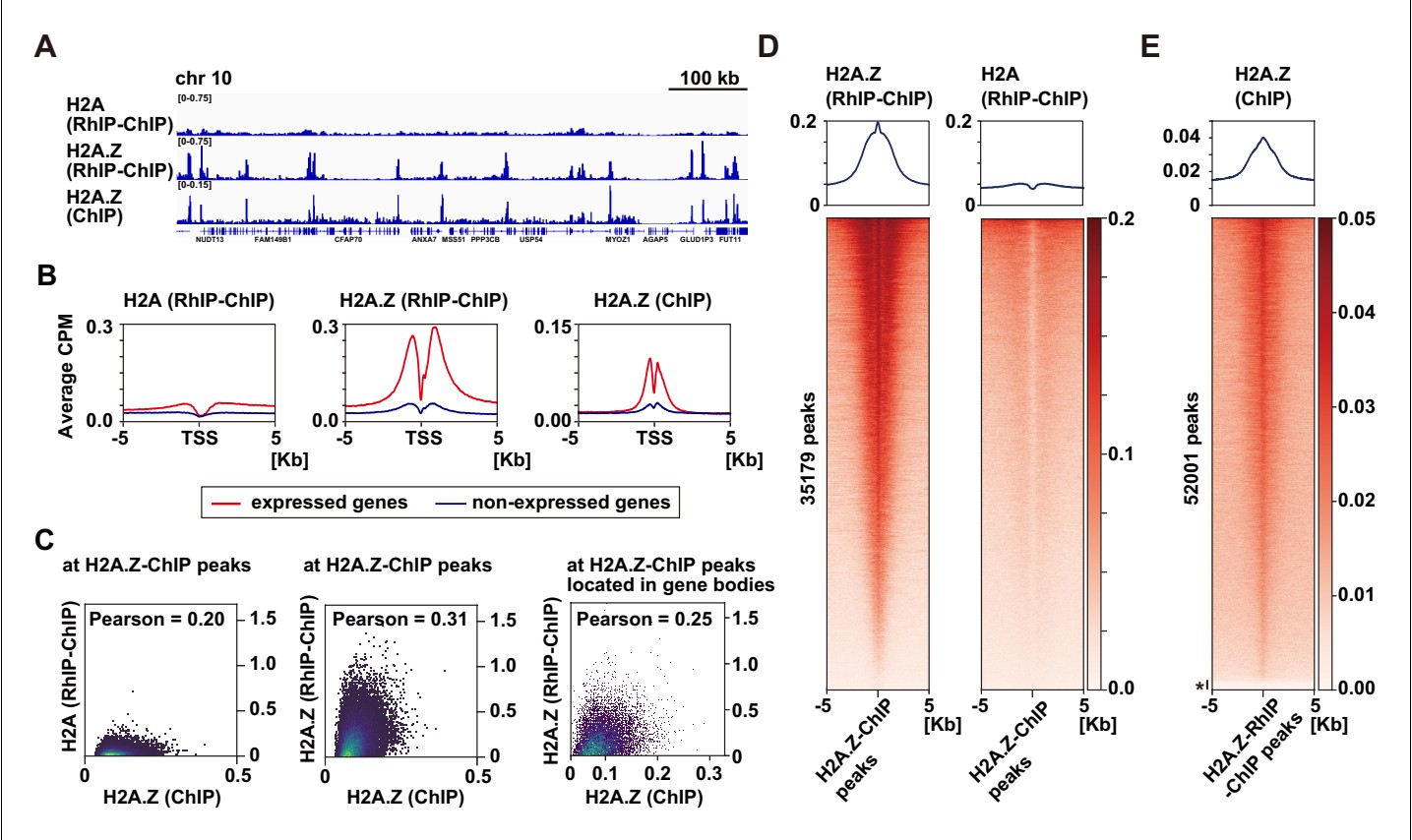

**Figure 4.** H2A.Z is specifically enriched around transcription start sites in the RhIP assay. (**A**) Representative profiles of RhIP-ChIP-seq using asynchronous cells and ChIP-seq (GEO:GSM1003483) at chr10: 73,079,443–73,805,171. (**B**) Aggregation plots of the H2A (RhIP-ChIP, left), H2A.Z (RhIP-ChIP, center), and H2A.Z (ChIP, right) at the center of TSS-flanking 5 Kb regions from the whole genome. Red and blue lines indicate expressed and non-expressed genes, respectively. (**C**) The correlations of H2A.Z (ChIP) and H2A (RhIP-ChIP) (left), and H2A.Z (ChIP), H2A.Z (RhIP-ChIP) (center) at known H2A.Z sites determined by ChIP-seq analysis. The correlations of H2A.Z (ChIP) and H2A.Z (RhIP-ChIP) at known H2A.Z sites located in gene bodies (right). (**D**) Heatmaps of H2A.Z and H2A (RhIP-ChIP) at the center of H2A.Z peaks of ChIP-seq flanking 5 Kb regions. The heatmap represents H2A.Z peaks ranked from the strongest to weakest in RhIP-ChIP-seq. Corresponding aggregation plots are at the top. (**E**) Reversed analysis of (**D**). Heatmap of H2A.Z (ChIP) at the H2A.Z peaks of RhIP-ChIP-seq. The corresponding aggregation plot is at the top. Asterisk indicates peaks with no read counts of H2A.Z (ChIP), which are approximately 3% of the entire peaks.

The online version of this article includes the following figure supplement(s) for figure 4:

**Figure supplement 1.** The biological replicate of the distribution analysis of incorporated H2A.Z in the RhIP-ChIP.

*Figure 4—figure supplement 1A*). In contrast, the H2A of RhIP-ChIP-seq showed no accumulation at the H2A.Z peaks of ChIP-seq (*Figure 4D* right). This is consistent with the results that the correlation of RhIP-ChIP-seq of H2A and H2A.Z at the H2A.Z peaks of ChIP-seq is lower than that along the genome (*Figure 3A* right and *Figure 3—figure supplement 2A* right). The reversed heatmap analysis again showed the accumulations of the H2A.Z of ChIP-seq at the H2A.Z peaks found by RhIP-ChIP-seq, and only 3% of the H2A.Z peaks of the RhIP assay were absent from those of ChIP-seq (denoted with * in *Figure 4E* and *Figure 4—figure supplement 1B*). These data revealed that the distribution of the incorporated H2A.Z in the RhIP assay overlapped well with the steady state localization of H2A.Z determined by the ChIP-seq analysis.

## H2A.Z deposition in the RhIP assay requires ANP32E and ATP supplements from the cellular extract

As H2A.Z is incorporated into the same regions of the endogenous H2A.Z in the RhIP assay, the preincorporated H2A.Z in the chromatin may be dynamically exchanged with the exogenously added H2A.Z in the RhIP assay. The exchange reaction requires the H2A.Z-H2B complex to be evicted from

nucleosomes, and ANP32E has this eviction activity (*Gursoy-Yuzugullu et al., 2015*; *Murphy et al., 2020*; *Murphy et al., 2018*; *Obri et al., 2014*). We then examined whether ANP32E is involved in the deposition of H2A.Z in the RhIP assay. Our fractionation analysis showed that ANP32E was extracted during the permeabilization process; in contrast, the other H2A.Z deposition factors SRCAP and TIP60 remained in the permeabilized cells (*Figure 5A*). Given that ANP32E exists in the cellular extract used in the RhIP assay (*Figure 5B*), we prepared cellular extracts from the control and ANP32E-knockdown cells (*Figure 5B*), and used them for the RhIP-ChIP-seq analysis of H2A.Z (*Figure 5C*). To allow a quantitative assessment of the ChIP-seq analysis, we combined spike-in controls with the RhIP-ChIP-seq (*Chen et al., 2015*; *Egan et al., 2016*; *Orlando et al., 2014*). The efficiencies of H2A.Z incorporations into active TSS and enhancer regions were decreased upon the ANP32E-knockdown (*Figure 5D* and *Figure 5—figure supplement 1A*). Previous reports showed that the endogenous H2A.Z is predominantly detected in TSS and to a lesser extent in enhancers (*Barski et al., 2007*; *Buschbeck and Hake, 2017*; *Ernst et al., 2011*; *Jin et al., 2009*; *Link et al., 2018*; *Obri et al., 2014*; *Schones et al., 2008*). Together with these observations, our results suggested that the ANP32E-dependent evictions of the pre-incorporated H2A.Z in active TSS and enhancer regions promote the incorporation of exogenously added H2A.Z.

In addition to eviction activity, chromatin remodeling is a key determinant for the H2A.Z localizations. The SRCAP and TIP60/EP400 complexes are chromatin remodeling factors that replace nucleosomal H2A with H2A.Z. These factors were found in the permeabilized cells, suggesting that chromatin remodeling is also involved in H2A.Z deposition in the RhIP assay (*Figure 5A*). A series of studies using the yeast SRCAP ortholog SWR1 revealed the mechanism of chromatin remodeling-mediated H2A.Z deposition and showed that ATPase activity is essential for the H2A.Z deposition (*Altaf et al., 2010*; *Luk et al., 2010*; *Mizuguchi, 2004*; *Sun and Luk, 2017*; *Sun et al., 2020*). Therefore, we investigated whether ATP is required for the deposition of H2A.Z in the RhIP assay. As cellular ATP is removed during the permeabilization process, the permeabilized cells lack ATP. We then prepared the ATP-reduced cellular extract and used it in a RhIP-ChIP-seq analysis of H2A.Z with or without ATP supplementation (*Figure 5E*). We found that ATP facilitated H2A.Z deposition at active TSS and enhancer regions as effectively as ANP32E (*Figure 5F* and *Figure 5—figure supplement 1B*). This indicated that the H2A.Z deposition in the RhIP assay is a net result of the ANP32E-mediated eviction and deposition of H2A.Z, and the replacement of pre-incorporated H2A with H2A.Z by chromatin remodeling. As ATP is used in multiple biological processes, such as transcription, splicing/translation, molecular chaperone functions, and so on, the possibility that these ATP-dependent reactions affected the H2A.Z incorporation in this assay cannot be excluded. Intriguingly, the efficiencies of H2A.Z incorporations into insulator regions are less affected under ANP32E- or ATP-reduced conditions (*Figure 5D and F*). This suggests that there may be another mechanism for H2A.Z deposition in insulator regions.

## The αc helix of H2A is important for its replication-coupled deposition

As the RhIP assay reproduces the replication-coupled and replication-independent incorporations of H2A and H2A.Z, we then tried to identify the residues responsible for the replication-coupled H2A and replication-independent H2A.Z depositions by a mutant analysis. A previous study showed that swapping the M6 region of H2A.Z with the corresponding H2A residues could not rescue the embryonic lethality of the H2A.Z null mutation in *Drosophila melanogaster* (*Clarkson et al., 1999*), suggesting that the region specifies the H2A.Z identity. The M6 region of H2A.Z and the corresponding region of H2A are exposed on the surface of the H2A.Z-H2B or H2A-H2B dimer (*Horikoshi et al., 2013*; *Luger et al., 1997*; *Suto et al., 2000*; *Tachiwana et al., 2010*; *Figure 6A*, cyan or green). This indicates that another protein can recognize the regions, which may be important for their depositions. To test this idea, we constructed the swapped mutant (H2A.Z_M6) and performed the RhIP assay (*Figure 6B–F*). Surprisingly, the H2A.Z_M6-H2B signals were observed in late replicating chromatin (*Figure 6D and E*), and its incorporation pattern in late S phase was more similar to that of H2A-H2B, rather than H2A.Z-H2B (*Figure 6F*). This indicated that the mutant is no longer H2A.Z, in terms of deposition. Thus, the M6 region of H2A.Z is responsible for the H2A.Z-specific deposition, and the corresponding region (amino acids 89–100) of H2A is responsible for the replication-coupled H2A deposition.

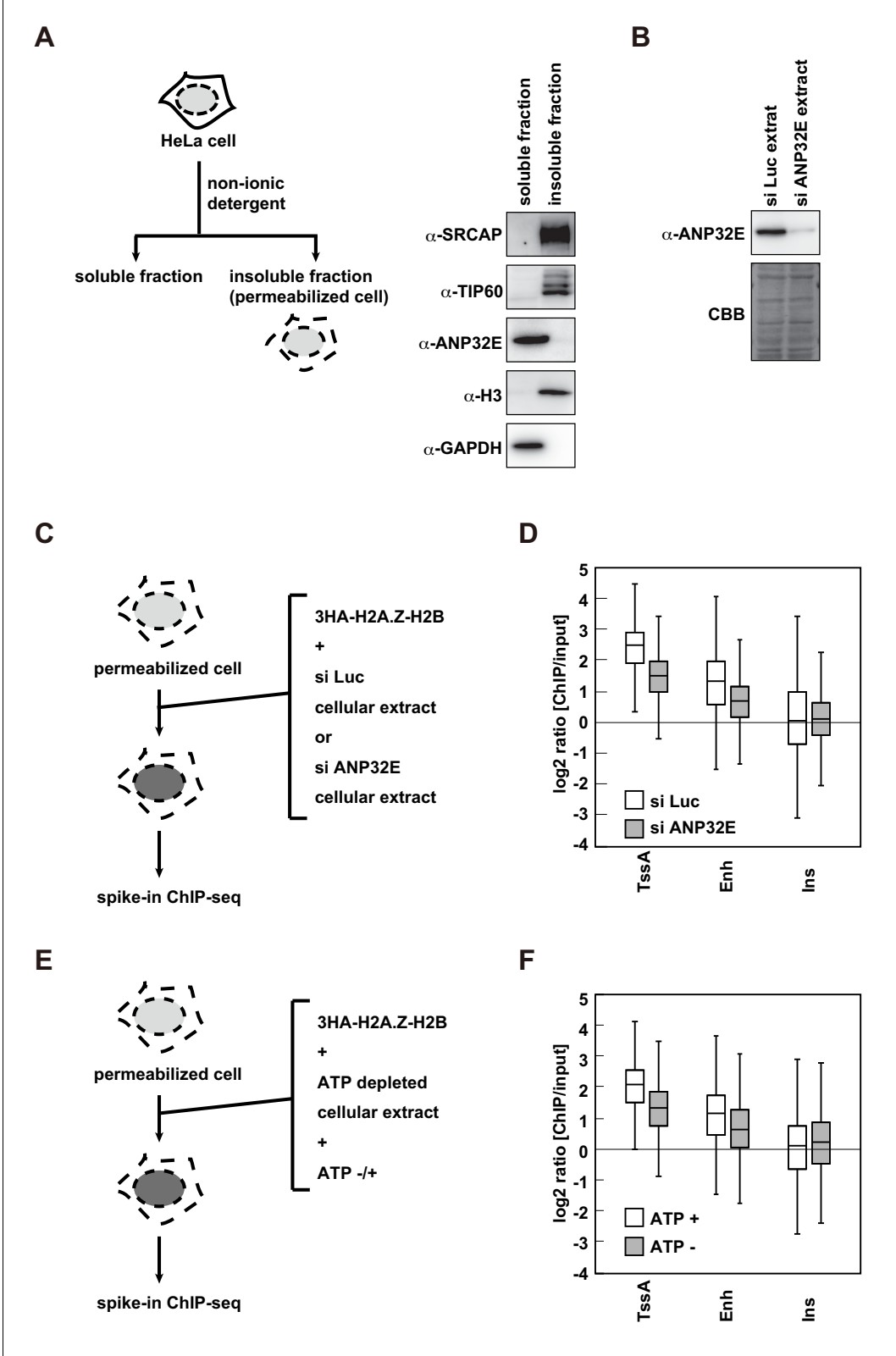

**Figure 5.** H2A.Z deposition requires both ANP32E and ATP in the RhIP assay. (**A**) Scheme of cell fractionation (left). HeLa cells were treated with non-ionic detergent and the supernatant (soluble fraction), which is removed from permeabilized cells, and the pellet, which corresponds to permeabilized cells, were separated (insoluble fraction). Distributions of SRCAP, TIP60 and ANP32E were investigated by western blotting (right). Histone H3 and

*Figure 5 continued on next page*

*Figure 5 continued*

GAPDH served as the chromatin and cytoplasmic protein controls, respectively. (**B**) Preparation of the ANP32E-knockdown cellular extract. HeLa cells were transfected with siRNA against ANP32E. After 72 hr, the ANP32E-knockdown cellular extract was prepared from the cells. The ANP32E-knockdown was confirmed by western blotting (upper). Loading control with the membrane stained with CBB (lower). (**C**) Scheme of RhIP-ChIP-seq of H2A.Z using the ANP32E-knockdown cellular extract. (**D**) Enrichment of incorporated H2A.Z with the control or ANP32E-knockdown extract. Each chromatin region was previously annotated by the chromHMM, as follows. TssA: active TSS, Enh: enhancers, and Ins: insulator. (**E**) Scheme of RhIP-ChIP-seq of H2A.Z using the ATP-depleted cellular extract. (**F**) Enrichment of incorporated H2A.Z with the ATP-depleted or ATP-supplemented extract. Each chromatin region was the same as in (**D**).

The online version of this article includes the following figure supplement(s) for figure 5:

**Figure supplement 1.** The biological replicate of RhIP-ChIP-seq analysis of H2A.Z using ANP32E- or ATP-depleted cellular extract.

## Discussion

We established the novel RhIP assay, combining permeabilized cells and reconstituted histone complexes, to analyze histone incorporation. Previous histone incorporation analyses using genetically encoded histone genes have revealed chromatin dynamics, including nucleosome turnover kinetics, but have limitations on the time resolution, as they require time to synthesize and/or label histones (*Deal and Henikoff, 2010*). In contrast, the RhIP assay can detect histone incorporation with better time resolution, as it does not require histone synthesis or labeling. In fact, we could analyze histone incorporations in the early and late S phases separately, using synchronized cells (*Figure 2—figure supplement 2* and *Figure 3D*). By combining RhIP with ChIP-seq, the RhIP assay enables the analysis of histone incorporation at the DNA sequence level, while ChIP-seq reveals their static presence. Scatter plot analyses of the H2A.Z RhIP-ChIP-seq replicates at known H2A.Z sites showed a quite high correlation (0.95 Pearson correlation coefficient), while the H2A and H2A.Z RhIP-ChIP-seq data showed a weaker correlation as compared with one of the replicates (*Figure 3—figure supplement 2A*). Thus, the RhIP-ChIP-seq analysis of H2A.Z also showed the reproducibility (*Figure 3*, *Figure 3—figure supplements 1* and *2*, *Figure 4* and *Figure 4—figure supplement 1*) and better enrichment (S/N ratio) as compared with ChIP-seq (*Figure 3—figure supplement 2B and C*), probably due to the high-affinity antibody against the epitope-tag. Together with the fact that the RhIP-ChIP assay requires a small number of cells (approximately $8 \times 10^6$ cells), the RhIP assay is suitable for the biochemical analysis of ChIP samples. The effects of post-translational modifications (PTMs) of histones on their incorporation have remained elusive. Methods to produce histones with PTMs in vitro have been developed (*Nadal et al., 2018*), and thus the analysis of the effects of PTMs on histone incorporation will be the next target for the RhIP assay.

The proper combination of histone chaperones and histone variants ensures correct histone localization. Therefore, if only histone variants are increased, then the excess histone variants may bind to unsuitable histone chaperones, leading to ectopic localization. Indeed, the overexpressed histone H3 variant, CENP-A, in cells aberrantly binds to the H3.3 chaperone, DAXX, leading to ectopic localizations (*Lacoste et al., 2014*). In the present study, the ectopic localizations of histones were not observed. Therefore, there may be sufficient amounts of free histone chaperones under these conditions. We performed the RhIP assay simultaneously with two different histone variants, which do not share the same chaperone. This suggests that they do not affect each other's incorporations. Since the amounts of exogenously added histones can be increased in the RhIP assay, it may be suitable for competition assays to analyze the overexpression of a specific histone variant.

The RhIP assay successfully reproduced the replication-coupled H3.1 incorporation and replication-independent H3.3 incorporation throughout the genome. Furthermore, the H3.1 and H3.3 incorporations required CAF-1 and HIRA, respectively, in this assay (*Figure 1* and *Figure 1—figure supplement 1*). Although CAF-1 is present in permeabilized cells, replication-coupled H3.1 deposition required the cellular extract (*Figure 1—figure supplement 2*). This suggests that one (or more) unknown cytosolic factor(s) couples newly synthesized H3.1 to chromatin-binding proteins, such as the CAF-1 complex.

The present study has revealed the correlation between histone incorporation and chromatin structure, using the RhIP assay. We analyzed the incorporation of the H2A family members, H2A,

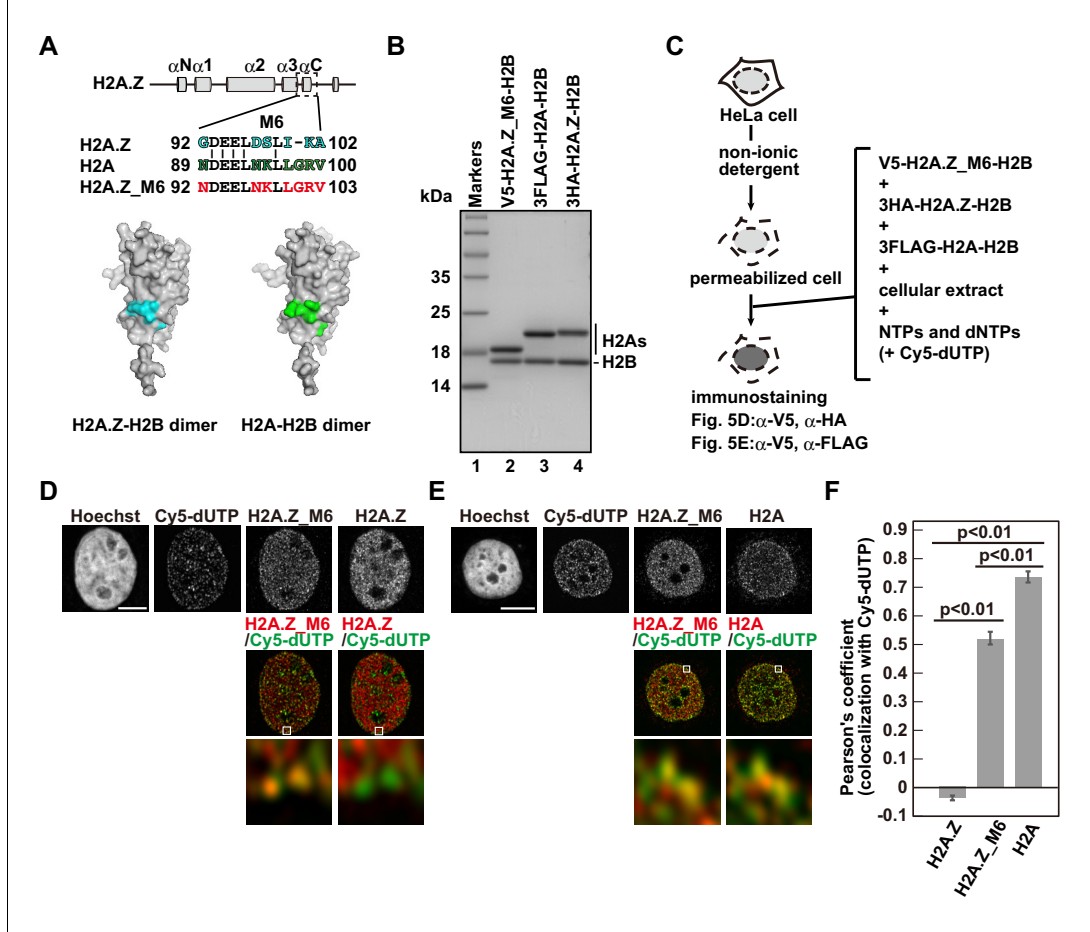

**Figure 6.** Identification of responsible residues for H2A- and H2A.Z-specific incorporations. (A) Amino acid alignments of the H2A.Z M6 region and its counterpart in H2A (upper). The structural models of the H2A.Z-H2B and H2A-H2B dimers (PDB IDs: 3WA9 and 3AFA, respectively). The specific residues are highlighted in cyan or green, respectively. All residues are located on the surface of the dimers. (B) Reconstituted H2AZ_M6-H2B, H2A-H2B, and H2A.Z-H2B complexes were analyzed by SDS-16% PAGE with Coomassie Brilliant Blue staining. H2AZ_M6, H2A, and H2A.Z were expressed as N-terminally V5, 3FLAG, and 3HA fused proteins, respectively. Lane 1 indicates the molecular mass markers, and lanes 2–4 indicate the H2A.Z_M6-H2B, H2A-H2B, and H2A.Z-H2B complexes, respectively. (C) Schematic representation of the RhIP assay, using the reconstituted H2A.Z_M6-H2B, H2A-H2B, and H2A.Z-H2B complexes. (D) RhIP-immunostaining images of H2A.Z and H2A.Z_M6. Exogenously added H2A.Z-H2B and H2A.Z_M6-H2B complexes were stained with the anti-V5 or -HA antibody. Cells in S phase were monitored with Cy5-dUTP. Middle: merged images of Cy5-dUTP (green) and H2A.Z_M6 (red) or H2A.Z (red). Bottom: magnified images of boxed areas are shown. Bar indicates 5 µm. (E) RhIP-immunostaining images of H2A and H2A.Z_M6. Exogenously added H2A-H2B and H2A.Z_M6-H2B complexes were stained with the anti-V5 or -FLAG antibody. Cells in S phase were monitored with Cy5-dUTP. Middle: merged images of Cy5-dUTP (green) and H2A.Z_M6 (red) or H2A (red). Bottom: magnified images of boxed areas are shown. Bar indicates 5 µm. (F) Colocalization analyses of Cy5-dUTP and H2A.Z, H2A.Z_M6 or H2A in late S phase (n > 35 cells). Experiments were repeated three times and averaged data are shown. The two-tailed Student's t-test was used for the statistical comparisons.

H2A.X, and H2A.Z. We found that the incorporations of H2A-H2B and H2A.X-H2B into transcriptionally active open chromatin can occur in both replication-coupled and -independent manners, whereas their incorporation into transcriptionally inactive closed chromatin occurs only in a replication-coupled manner in late S phase (*Figures 2*, *3* and *6*). This indicated that histone exchange would rarely occur in closed chromatin. Thus, the genome-wide localization of H2A and H2A.X is achieved by a combination of deposition into open chromatin and replication-coupled deposition into closed chromatin (*Figure 7*). In contrast, H2A.Z exhibited a much lower frequency of replication-coupled deposition, as compared to H2A (*Figure 2F* and *Figure 2—figure supplement 2*). Together with the fact that the amount of H2A.Z is much lower than that of H2A in S phase cells (*Wu et al., 1982*), we concluded that little to no H2A.Z is incorporated into closed chromatin (*Figure 7*). This is consistent with previous observations that the H2A.Z and DNA methylation localizations are mutually exclusive (*Nothjunge et al., 2017*; *Zilberman et al., 2008*), and that H2A.Z predominantly exists at

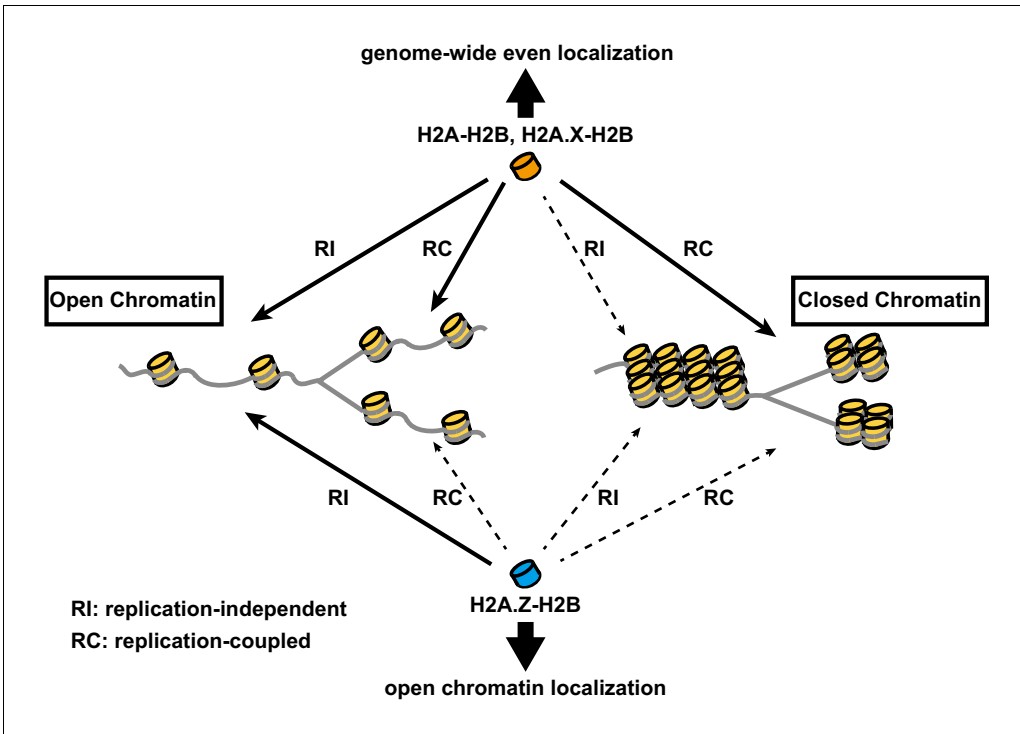

**Figure 7.** Model of differential histone incorporations into open and closed chromatin. In open chromatin, the H2A-H2B and H2A.X-H2B complexes are incorporated in replication-independent (RI) and replication-coupled (RC) manners, respectively, while H2A.Z-H2B is incorporated only in a replication-independent manner. In closed chromatin, new histone depositions of H2A and H2A.X, but not H2A.Z, occur only in a replication-coupled manner. This leads to the global localizations of H2A-H2B and H2A.X-H2B, as well as the specific localization of H2A.Z-H2B, including its elimination from closed chromatin.

promoters and enhancers (*Barski et al., 2007*; *Buschbeck and Hake, 2017*; *Ernst et al., 2011*; *Jin et al., 2009*; *Link et al., 2018*; *Obri et al., 2014*; *Schones et al., 2008*). The means by which H2A.Z becomes enriched at specific regions of open chromatin have remained enigmatic. Our study suggested that the H2A.Z elimination from transcriptionally inactive chromatin is due to the low frequencies of histone exchange and replication-coupled H2A.Z incorporation, which may partially explain the H2A.Z distribution pattern (*Figure 7*). Moreover, H2A.Z was specifically incorporated into chromatin around the TSS of expressed genes in the RhIP assay, in excellent agreement with the steady state localization of H2A.Z revealed by ChIP-seq (*ENCODE Project Consortium, 2012*). We also found that the H2A.Z depositions in active TSS and enhancer regions decreased in the ANP32E knockdown cellular extract, indicating that the ANP32E-mediated eviction promotes the new deposition of H2A.Z in the RhIP assay. However, other mechanism(s) may also exist, because the decrease in the H2A.Z deposition was relatively moderate (*Figure 5C and D*). This is consistent with the results that ANP32E knockout mice did not show any apparent effects on their viability (*Reilly et al., 2010*), even though proper H2A.Z incorporation is essential for cell survival and proliferation. Our data also showed that ATP promoted the H2A.Z deposition in active TSS and enhancer regions (*Figure 5E and F*), suggesting that the H2A.Z deposition is an ATP-dependent process. Taken together, the H2A.Z deposition in the RhIP assay is the net result of the ANP32E-mediated eviction and deposition of H2A.Z, and the replacement of pre-incorporated H2A with H2A.Z by ATP-dependent processes. We also found that the correlation between the RhIP-ChIP-seq and ChIP-seq of H2A.Z at known H2A.Z peaks in the gene body is lower, as compared with all known H2A.Z peaks (*Figure 4C* center and right). This may reflect the fact that the RhIP assay cannot assess histone recycling; That is, the re-deposition of the evicted histone, which is observed during transcription (*Jeronimo et al., 2019*; *Torné et al., 2020*). H2A.Z is also found at pericentric heterochromatin (*Boyarchuk et al., 2014*; *Greaves et al., 2007*; *Rangasamy et al., 2003*). As human pericentric

heterochromatin is composed of repetitive DNA sequences (satellite repeats), we could not analyze H2A.Z incorporation into pericentric heterochromatin by the RhIP-ChIP-seq assay (*Figure 3*). Previous studies showed that the transcriptional activation of pericentric satellites accompanies the structural changes from heterochromatin to euchromatin (*Jolly et al., 2004*; *Rizzi et al., 2004*; *Saksouk et al., 2015*; *Valgardsdottir et al., 2005*). In addition, the pericentric region of human chromosome nine is heterogeneous, with both heterochromatic and euchromatic regions (*Gilbert et al., 2004*). Therefore, H2A.Z incorporation into pericentric heterochromatin may occur, as in the open chromatin of the region.

We found that histone incorporations are regulated by the chromatin structure, which may be important for maintaining the closed chromatin configuration. Histones reportedly form a pre-deposition complex, which includes many transcription-related factors, before their incorporation into chromatin in vivo (*Dunleavy et al., 2009*; *Mao et al., 2014*; *Obri et al., 2014*; *Tagami et al., 2004*). For instance, the H2A-H2B pre-deposition complex contains Spt16 and SSRP1, which form the heterodimer complex FACT that functions in transcription facilitation, and the pre-deposition H2A.Z-H2B complex also includes a chromatin remodeling factor (SRCAP), a histone acetyltransferase (TIP60), and acetyl-lysine-binding proteins (GAS41, Brd8). If histone exchange usually occurs in closed chromatin, then these transcription-related factors might accumulate in the closed chromatin and alter the epigenetic chromatin states. Therefore, the deficiency of replication-independent histone exchange in closed chromatin may be important for maintaining a transcriptionally inactive state.

In spite of the high-sequence homology between H2A and H2A.Z, their localizations are different. By using the swapping mutant, we analyzed the residues responsible for the specific depositions of H2A and H2A.Z (*Figure 6*). We identified residues 88–100 of H2A as being responsible for its replication-coupled deposition. Although the means by which this region contributes to the incorporation into replicating chromatin remain unknown, a factor that binds this region and allows H2A to assemble at a replicating site may exist. The major H2A-specific chaperones, Spt16 and Nap1, which are components of the H2A pre-deposition complex, do not bind to this region (*Aguilar-Gurrieri et al., 2016*; *Kemble et al., 2015*). Thus, the protein that recognizes these residues is likely to be a chromatin protein involved in replication, rather than a component of the H2A pre-deposition complex. This region is a counterpart of the H2A.Z M6 region, and the H2A.Z_M6 swapping mutant did not compensate for the embryonic lethality of the H2A.Z knockout in *Drosophila melanogaster* (*Clarkson et al., 1999*). As this region is essential for the H2A.Z eviction by ANP32E (*Gursoy-Yuzugullu et al., 2015*; *Obri et al., 2014*), the incorporated H2A.Z_M6 mutant may not be removed from closed chromatin, thus resulting in aberrant gene expression and impaired embryonic development.

In conclusion, our novel method elucidated the mechanism of histone incorporation at the DNA sequence level, and revealed that the chromatin structure is the first determinant of histone localization.

## Materials and methods

### Key resources table

| Reagent type (species) or resource | Designation | Source or reference | Identifiers | Additional information |
|---|---|---|---|---|
| Antibody | Anti-HA (mouse, monoclonal) | Santa Cruz | sc-7392 RRID:AB_627809 | IF(1:1,000) |
| Antibody | Anti-HA (rabbit, monoclonal) | Cell Signaling Technology | 3724 RRID:AB_1549585 | IF(1:2,000) only used in *Figure 1—figure supplements 1D* and *3D* |
| Antibody | Anti-DDDDK (anti-FLAG) (rabbit, polyclonal) | MBL | PM020 RRID:AB_591224 | IF(1:500) |
| Antibody | Anti-V5 (chicken, polyclonal) | Abcam | ab9113 RRID:AB_307022 | IF(1:1000) |
| Antibody | Anti-p60 (rabbit, monoclonal) | Abcam | ab109442 RRID:AB_10861771 | IF(1:400) |

*Continued on next page*

*Continued*

| Reagent type (species) or resource | Designation | Source or reference | Identifiers | Additional information |
|---|---|---|---|---|
| Antibody | Anti-HIRA (mouse, monoclonal) | Active Motif | WC119.2H11 RRID:AB_10715607 | IF(1:200) |
| Antibody | Anti-HIRA (rabbit, monoclonal) | Abcam | ab129169 RRID:AB_11140220 | WB(1:500) |
| Antibody | Anti-p60 (rabbit, monoclonal) | Abcam | ab109442 RRID:AB_10861771 | WB(1:400) |
| Antibody | Anti-H3 (mouse, monoclonal) | MBL | MABI0301 RRID:AB_11142498 | WB(1:1000) |
| Antibody | Anti-SRCAP (rabbit, polyclonal) | Kerafast | ESL103 | WB(1:1000) |
| Antibody | Anti-TIP60 (mouse, monoclonal) | Santa Cruz | sc-166323 RRID:AB_2296327 | WB(1:50) |
| Antibody | Anti-ANP32E (rabbit, polyclonal) | MyBioSource | MBS9214243 | WB(1:1000) |
| Antibody | Anti-GAPDH (mouse, monoclonal) | MBL | ML171-3 RRID:AB_10597731 | WB(3:1000) |
| Sequence-based reagent | GAPDH_F | This paper | PCR primers | AAAGGGTGCAGCTGAGCTAG |
| Sequence-based reagent | GAPDH_R | This paper | PCR primers | TACGAAGCCCTTCCAGGAGA |
| Sequence-based reagent | LINC02199_F | This paper | PCR primers | CCGGTGTCAAATGTCACAATGAA |
| Sequence-based reagent | LINC02199_R | Thermo Fisher | PCR primers | GGGGTTTTGAGGATTCCAAAGTG |
| Sequence-based reagent | si_p60 gene | This paper | siRNA | AAUGAUAACAAGGAGCCGGAGdTdT |
| Sequence-based reagent | si_p150 gene | This paper | siRNA | CUGUCAUGUGGGUUCUGACdTdT |
| Sequence-based reagent | ON-TARGET SMARTpool siRNA (HIRA gene) | Dharmacon | siRNA | L-013610-00-0005 |
| Cell line (*Homo sapiens*) | HeLa | Peter R Cook | N/A | A human cervical cancer cell line (female origin) |
| Cell line (*Homo sapiens*) | SF8628 | Merck | SCC127 RRID:CVCL_IT46 | Human DIPG H3.3-K27M Cell Line |

## Cell culture and thymidine block

HeLa and SF8628 cells were cultured in DMEM supplemented with 10% fetal bovine serum, at 37°C in a 5% $CO_2$ atmosphere. For cell synchronization, HeLa cells were cultured with 2 mM thymidine for 18 hr (*Figure 1—figure supplement 2C*). For the double thymidine block, the medium was then changed to remove the thymidine. After 9 hr of culture without thymidine, the HeLa cells were cultured with 2 mM thymidine again for 15 hr, for synchronization in S phase (*Figure 2—figure supplement 2*). All cell lines used in this study had their identities validated by STR profiling and it was also confirmed by Hoechst staining that there was no mycoplasma contamination.

## Reconstitution of histone complex

All histones used in this study were produced in *Escherichia coli* cells as recombinant proteins. The human H3.1 and H3.3 genes were inserted in the pET21a vector (Novagen), as C-terminal 3HA-His$_6$ or 3FLAG-His$_6$ fused genes. The human H2A, H2A.X, H2A.Z, and H2A.Z_M6 genes were inserted in the pET15b vector (Novagen) as N-terminal His$_6$-3HA, His$_6$-3FLAG or His$_6$-V5 fused genes. All the genes were overexpressed in the BL21(DE3) *E. coli* strain, by adding 0.5 mM isopropyl-β-D-thiogalactopyranoside. Each histone was then purified as described previously, using Ni-NTA affinity chromatography (*Tachiwana et al., 2010*). The epitope tag-fused histones were freeze-dried without removing the His$_6$ tags. As the His$_6$ tag was not used to detect or precipitate histones in this study,

His$_6$ was not mentioned in the text and figures, so that His$_6$-epitope-tag-histone and histone-epitope-tag-His$_6$ were referred to as epitope-tag-histone and histone-epitope-tag, respectively. Human H2B and H4 were overexpressed and purified after removing the His$_6$ tags, as described previously (*Tachiwana et al., 2010*).

Freeze-dried H3.1 or H3.3 was mixed with H4 and H2A, H2A.X, H2A.Z, and H2A.Z_M6, along with H2B, in 20 mM Tris-HCl buffer (pH 7.5), containing 7 M guanidine hydrochloride and 20 mM 2-mercaptoethanol, and incubated on ice. After 1 hr, the samples were dialyzed against reconstitution buffer (10 mM Tris-HCl, pH 7.5, and 2 mM 2-mercaptoethanol) containing 2 M NaCl, overnight at 4°C. The NaCl concentration was then decreased by three steps of dialysis against reconstitution buffer containing 1 M NaCl for 4 hr, 0.5 M NaCl for 4 hr, and 0.1 M NaCl overnight. After the dialyses, the precipitants were removed by centrifugation and the supernatants were analyzed by Superdex 200 gel filtration chromatography (GE Healthcare).

## Preparation of the cellular extract

The cellular extract was prepared as previously described, with modifications (*Martini et al., 1998*). Confluent HeLa cells in twenty-five 15 cm dishes were rinsed twice with ice-cold PBS in a cold room. Then, 10 ml of ice-cold hypotonic buffer (10 mM HEPES-KOH, pH 7.8, 10 mM KCL, 1.5 mM MgCl$_2$, 1 mM dithiothreitol and 1 × proteinase inhibitor cocktail [cOmplete, EDTA-free, Roche]) was added to each dish to swell the cells, and the dishes were incubated for 5 min in a cold room. After repeating this step once, the excess buffer was removed by leaning the dishes against a wall for 5 min. The cells were then scraped off the dish and disrupted with 35 strokes in a 1 ml Dounce homogenizer with a loose-fitting pestle (Wheaton). Nuclei were removed by centrifugation at 1,500 × g for 5 min at 4°C, and the collected supernatant was further centrifuged at 14,000 × g for 5 min at 4°C. The supernatant was then flash-frozen with liquid nitrogen. After 30 min at −80°C, the cellular extract was thawed and the debris was removed. The concentration of the cellular extract was then measured with a protein quantification kit (MACHEREY-NAGEL). Usually, 3–4 ml of 5–8 mg/ml cellular extract were obtained.

For the preparation of the ATP-depleted cellular extract, HeLa cells were grown to confluence in a 15 cm dish, and further incubated for 60 min at 37°C in DMEM without glucose and fetal bovine serum, supplemented with 6 mM 2-deoxy-D-glucose and 10 mM sodium azide (*Schwoebel et al., 2002*). The cellular extract was then prepared as described above. Since the ATP-depleted cells were prone to detachment, all steps were performed gently.

For the preparation of the knockdown cellular extract, HeLa cells were transfected with siRNAs against the luciferase or ANP32E gene using the Lipofectamine RNAiMAX Transfection Reagent (Thermo Fisher), according to the manufacturer's instructions. After 72 hr, the cellular extract was prepared as described above. The siRNAs used are as follows: siRNA against luciferase gene: CGUACGCGGAAUACUUCGAdTdT siRNA against ANP32E gene: AUGGAUUUGAUCAGGAGGAdTdT.

## RhIP-immunostaining

HeLa cells were grown on a coverslip in a six well dish to 2.5 × 10$^5$ per well. The cells were chilled on ice and rinsed twice with 2 ml of ice-cold PBF (100 mM CH$_3$COOK, 10 mM Na$_2$HPO$_4$, 30 mM KCl, 1 mM dithiothreitol, 1 mM MgCl$_2$, 1 mM ATP, and 5% Ficoll). To permeabilize the cells, 1 ml of PBF containing 0.2% Triton X-100 was added, and the cells were incubated for 5 min on ice and then rinsed twice with 2 ml of ice-cold PBF. As the permeabilized cells were easily detached from the coverslips, all subsequent steps were performed gently. The coverslips with the attached permeabilized cells were moved onto parafilm laid on an aluminum block, and incubated at 30°C. Then, a 50 µl portion of a RhIP-reaction mixture, containing 100 nM histone complex, 3.6 µg/µl cellular extract, 2.5% Ficoll, 100 mM CH$_3$COOK, 10 mM Na$_2$HPO$_4$, 30 mM KCl, 1 mM dithiothreitol, 1 mM MgCl$_2$, 100 µM each of dNTPs, and NTPs (Roche) with or without 250 nM Cy5-dUTP (Enzo Life Sciences), was added to the permeabilized cells, and incubated for 60 min at 30°C. The coverslip was placed in a well of a 12-well plate, and the cells were washed twice for 5 min with 1 ml of PBS containing 0.1% Tween 20 (PBST), at room temperature. The cells were then fixed with 4% PFA (Electron Microscopy Sciences) in PBS for 20 min at room temperature, and rinsed three times with PBS. The cells were treated with 1% BSA in PBST for 1 hr at room temperature, and then incubated for 2 hr at room temperature with one of the following primary antibodies: anti-HA (mouse, 1/1000, Santa

Cruz, sc-7392), anti-HA (only used in *Figure 1—figure supplements 1D* and *3D*, rabbit, 1/2,000, Cell Signaling Technology, 3724), anti-DDDDK (FLAG) (rabbit, 1/500, MBL, PM020), anti-V5 (chicken, 1/1000, Abcam, ab9113), anti-p60 (rabbit, 1/400, Abcam, ab109442), and anti-HIRA (mouse, 1/200, Active Motif, WC119.2H11). The cells were washed with PBST three times for 10 min each, and then incubated for 1 hr at room temperature with secondary antibodies: Goat Alexa Fluor 488 or 546-conjugated anti-mouse IgG (1/1000, Life Technologies), or goat DyLight 488- or 550-conjugated anti-rabbit IgG or chicken IgY (1/1000, Thermo Fisher). The cells were washed with PBST three times for 10 min each. DNA was stained with Hoechst 33342. The samples were mounted with ProLong Gold (Life Technologies). The images in *Figure 1*, *Figure 1—figure supplements 2* and *3*, and *Figure 2—figure supplements 1* and *2* were acquired by using the Deltavision set-up (Cytiva) with an inverted Olympus IX71 microscope, equipped with a CoolSNAP ES2 CCD camera (Photometrics) and a 60×, 1.42 Plan Apo N Olympus oil-immersion objective. Other images were acquired with an LSM 880 inverted confocal microscope (Zeiss), equipped with an AiryScan module and a 63×, 1.40 Plan-Apochromat Zeiss oil objective. All image files were converted to the TIFF format using the ImageJ software (*Schneider et al., 2012*) and imported into Illustrator (Adobe) for assembly. The co-localization analysis was performed using the ImageJ Colocalization_Finder plugin.

For the RhIP-immunostaining assay with the CAF-1- or HIRA-knockdown, HeLa cells were transfected with siRNAs against the luciferase (negative control), CAF-1 (p60), CAF-1 (p150), or HIRA gene using the Lipofectamine RNAiMAX Transfection Reagent (ThermoFisher), according to the manufacturer's instructions. To knockdown the CAF-1 complex, the siRNAs against the CAF-1-p60 and CAF-I-p150 genes were added simultaneously. After an incubation for 8 hr, the cells were collected by trypsin/EDTA treatment, and the CAF-1- or HIRA-knockdown cells were mixed with control-knockdown cells and re-plated and co-cultured for 64 hr. For the CAF-1-knockdown experiment, 2 mM of thymidine was added to synchronize the cells in S phase, 18 hr before the permeabilization. Immunostainings were then performed as described above. siRNAs used for this experiment are as follows: siRNA against p60 gene: AAUGAUAACAAGGAGCCGGAGdTdT siRNA against p150 gene: CUGUCAUGUGGGUUCUGACdTdT siRNA against HIRA gene: ON-TARGET SMARTpool siRNA L-013610-00-0005 (Dharmacon).

## RhIP-ChIP

The RhIP-ChIP assays shown in *Figures 1H* and *2F* and *Figure 2—figure supplement 2* were performed as follows. Semi-confluent HeLa cells in a 10 cm dish were chilled on ice, and rinsed twice with 5 ml of ice-cold PBF. To permeabilize the cells, 2.5 ml of PBF containing 0.2% Triton X-100 was added and incubated for 5 min on ice. The cells were rinsed twice with 5 ml of ice-cold PBF, and then 2.5 ml of the RhIP-reaction mixture was added to the permeabilized cells. After sealing the lid of the dish with parafilm, the cells were incubated for 60 min at 30°C. The RhIP-reaction mixture was then removed, and the cells were washed twice with 5 ml of PBST at room temperature for 5 min each. Subsequently, native-ChIP was performed as described previously with minor modifications (*Tachiwana et al., 2015*). The cells were collected in 1 ml of NB buffer (15 mM Tris-HCl, pH 7.5, 15 mM NaCl, 60 mM KCl, 300 mM sucrose and 1 × proteinase inhibitor cocktail (cOmplete, EDTA-free, Roche)). After centrifugation at 1500 × g for 5 min at 4°C, the cells were resuspended in 100 µl of NB buffer. $CaCl_2$ was added to a final concentration of 2 mM. The cells were treated with 150 mU/µl of micrococcal nuclease (MNase, Worthington) for 5 min at 37°C, to generate soluble chromatin fragments. The reaction was terminated by adding 10 mM EDTA, and the solubilized chromatin fragments were separated from the pellets by centrifugation at 16,000 × g for 5 min at 4°C. The samples were then mixed with 50 µl of anti-HA-tag mAb-Magnetic Beads (MBL International) in 15 mM Tris-HCl, pH 7.5, 300 mM NaCl, and 0.1% NP-40. After an overnight incubation at 4°C with gentle mixing on a wheel, the beads were washed three times each with 500 µl of ChIP buffer (10 mM Tris-HCl, pH 8.0, 200 mM KCl, 1 mM $CaCl_2$, 0.5% NP-40), ChIP buffer containing 500 mM KCl, and TE. The DNA was then eluted with a Proteinase K solution (20 mM Tris-HCl, pH 8.0, 20 mM EDTA, 0.5% sodium dodecyl sulfate (SDS), and 0.5 mg/ml Proteinase K) and extracted with phenol-chloroform. The DNA was precipitated with ethanol and resuspended in 10 mM Tris-HCl, pH 8.0, and 1 mM EDTA. The resulting DNA samples were separated by electrophoresis on a 2% agarose gel (7.4 V/cm, 35 min) (*Figures 1H* and *2F*) or a 1.5% agarose gel (7.1 V/cm, 60 min) (*Figure 2—figure supplement 2B*) in 1 × TAE, and stained with SYBR Gold (Thermo Fisher). Images of the DNA stained

with SYBR Gold and the nascent DNA labeled with Cy5 were obtained with an Amersham Typhoon scanner (Cytiva).

The H3.3 enrichment shown in *Figure 1E* was analyzed by qPCR with a StepOne Plus system (Applied Biosystems) using SYBR Green fluorescence. The cycle number required to reach the threshold was recorded and analyzed. PCR was performed using the same amounts of the immuno-precipitated DNA (RhIP-ChIP sample) and the input DNA. Values were normalized to the input DNA. Primer sets used for this analysis are as follows:

> GAPDH forward: AAAGGGTGCAGCTGAGCTAG
> GAPDH reverse: TACGAAGCCCTTCCAGGAGA
> LINC02199 forward: CCGGTGTCAAATGTCACAATGAA
> LINC02199 reverse: GGGGTTTTGAGGATTCCAAAGTG.

## RhIP-ChIP-seq

RhIP-ChIP-seq was performed in the same way as RhIP-ChIP with modifications. HeLa cells were used in *Figures 3–5* and SF2868 cells were used in *Figure 1—figure supplement 1A*. After the RhIP assay, the permeabilized cells were washed twice with PBST for 5 min at room temperature, and then cross-linked with 3% formaldehyde in PBS for 5 min at room temperature. After fixation, the cells were resuspended with 1 ml of ChIP buffer and collected by centrifugation at 1500 × g for 5 min at 4°C. The cells were then treated with 7.5 mU/μl MNase in 400 μl ChIP buffer for 30 min at 37°C. The MNase reaction was terminated by adding 10 mM EDTA, and the solubilized chromatin fragments were separated from the pellets by centrifugation at 16,000 × g for 5 min at 4°C. The supernatant was incubated with 50 μl of anti-HA-tag mAb-Magnetic Beads, and gently mixed by rotation at 4°C overnight. The beads were then washed three times each with 500 μl of ChIP buffer, ChIP buffer containing 500 mM KCl, and TE. The beads were resuspended in 100 μl of ChIP elution buffer (50 mM Tris-HCl, pH 8.0, 10 mM EDTA, 25 mM NaCl, and 1% SDS), and incubated overnight at 65°C to reverse the cross-links. The DNA was then eluted with 0.4 mg/ml Proteinase K, and further purified with a PCR clean-up kit (MACHEREY-NAGEL). The DNA libraries were prepared using a SMARTer ThruPLEX Tag-seq Kit (Takara Bio) for *Figures 3* and *4*, or a SMARTer ThruPLEX DNA-seq kit (Takara Bio) for *Figure 5*. The samples were sequenced on an Illumina HiSeq 1500 system for *Figures 3* and *4*, and an Illumina HiSeq X Ten system for *Figure 5*.

For the spike-in normalization performed in *Figure 5*, one-tenth of the fragmented chromatin was subjected to cross-link reversal prior to the ChIP procedure. After an overnight incubation at 65°C, the resulting DNA was recovered as described above and the concentration was measured. Then, one two-hundredth of *Drosophila* spike-in chromatin (Active Motif) was mixed with the rest of the chromatin. Two μg of spike-in antibody (Active Motif) per 50 ng of *Drosophila* spike-in chromatin were mixed with ProteinA/G magnetic beads (Pierce). The spike-in antibody beads were then mixed with 50 μl of anti-HA-tag mAb-Magnetic Beads and the chromatin mixture. Subsequent steps were performed as described above.

## ChIP-seq data analysis

For ChIP-seq data shown in *Figures 3* and *4*, the in-read unique-molecular-identifiers (tags) were extracted using UMI-tools (*Smith et al., 2017*) with the command: umi_tools extract —`extract-method=regex` —`bc-pattern`='(?p<umi_1>.{6})(?p<discard_1>.{0,3}GTAGCTCA){s <= 2}'. The extracted reads were mapped to the human genome (GRCh38) using hisat2 (version 2.1.0) (*Kim et al., 2015*). The PCR-duplicates were removed using umi_tools dedup. The read counts on 15 chromatin states were calculated using BED-Tools (*Quinlan and Hall, 2010*). The definition of the ChromHMM track was obtained from the consolidated data set of the Roadmap Epigenomics project (E117_15_coreMarks_hg38lift_mnemonics.bed) (*Roadmap Epigenomics Consortium et al., 2015*). The BED files of enhancer and insulator regions used in *Figure 5* were also extracted from ChromHMM track. The overall concentrations of the ChIP signals (log2 ratio) were calculated as the ratio of the proportion of reads in each chromatin state between the ChIP and input DNA data; that is, $\log_2$(ChIP/Input) after normalization of the total reads. The signal tracks (bigwig files) were created at 1 bp intervals on the genome, and then the counts were normalized as CPM (Reads Per Million reads), using deepTools (*Ramírez et al., 2014*). ChIP-seq signals were visualized with the Integrative Genomics Viewer, IGV (*Robinson et al., 2011*). The Pearson correlation coefficients were

calculated throughout 800 bp intervals with the multiBigwigSummary program of deepTools, and plotted with the plotCorrelation program (*Ramírez et al., 2014*). The multiBigwigSummary program was also used to compute the average scores for RhIP-ChIP-seq data and ChIP-seq data (GEO: GSM1003483) at H2A.Z sites, which were determined with replicates of the ChIP-seq data (GEO: GSM1003483) by the mergePeaks program of HOMER (*Heinz et al., 2010*). The computeMatrix programs of deepTools were utilized to analyze the peak localizations at the centers of TSS, the known H2A.Z sites, and the H2A.Z peaks of the RhIP-ChIP-seq data. Genes with five or more read counts in the RNA-seq data (GSE140768 for *Figure 1—figure supplement 1*, GSE123571 for *Figure 4B*) were defined as expressed genes, and the rest were regarded as non-expressed genes. The H2A.Z peaks of the RhIP-ChIP-seq data were also determined with the mergePeaks program of HOMER. The plotFingerprint program of deepTools was utilized to analyze the enrichment of ChIP signals.

For the spike-in normalization used in *Figure 5*, the reads were mapped to the human genome (GRCh38) and the fly genome (BDGP Release 6 + ISO1 MT/dm6), using bowtie2 (version 2.3.4.3) (*Langmead and Salzberg, 2012*). The PCR-duplicates were removed using the MarkDuplicates tool of Picard (*Broad Institute, 2019*). Normalization factors were then generated using the numbers of mapped reads on the fly genome. The numbers of mapped reads on the human genome were then downsampled, using the view program of SAMtools according to the normalization factors (*1000 Genome Project Data Processing Subgroup et al., 2009*).

## Purification of NAP1 and ASF1

Human NAP1 was purified as described previously (*Tachiwana et al., 2008*). The human ASF1 gene was inserted in the pET15b vector (Novagen), in which the thrombin proteinase recognition sequence was replaced by the PreScission protease recognition sequence. *Escherichia coli* strain BL21-CodonPlus (DE3)-RIL cells (Agilent Technologies) were freshly transformed with the vector, and cultured at 30℃. After the cell density reached an $A_{600}$ = 0.8, 1 mM isopropyl- β-D-thiogalactopyranoside was added, and the culture was continued at 18℃ for 12 hr to induce His$_6$-tagged Asf1 expression. The cells were collected and resuspended in 20 mM Tris-HCl, pH 7.5, containing 500 mM KCl, 10% glycerol, 0.1% NP-40, 1 mM phenylmethylsulfonyl fluoride (PMSF), and 2 mM 2-mercaptoethanol. After cell disruption by sonication, the debris was removed by centrifugation (27,216 × g; 20 min), and the clarified lysate was mixed gently with 4 ml (50% slurry) of nickel-nitrilotriacetic acid (Ni-NTA)-agarose resin (Qiagen) at 4℃ for 1 hr. The Ni-NTA beads were washed with 200 ml of 20 mM Tris-HCl, pH 7.5, containing 500 mM NaCl, 10% glycerol, 10 mM imidazole, and 2 mM 2-mercaptoethanol. The His$_6$-tagged Asf1 was eluted by a 100 ml linear gradient of 10 to 500 mM imidazole in 50 mM Tris-HCl buffer (pH 7.5), containing 500 mM NaCl, 10% glycerol, and 2 mM 2-mercaptoethanol. PreScission protease (8 units/mg protein, Cytiva) was added to remove the His$_6$ tag from the ASF1. The sample was dialyzed against 20 mM Tris-HCl, pH 7.5, containing 100 mM NaCl, 1 mM EDTA, 10% glycerol, and 2 mM 2-mercaptoethanol. The ASF1 was further purified by chromatography on a Mono Q (Cytiva) column, eluted with a 25 ml linear gradient of 100–600 mM NaCl in 20 mM Tris-HCl, pH 7.5, containing 1 mM EDTA, 10% glycerol, and 2 mM 2-mercaptoethanol. The eluted Asf1 was further purified by chromatography on a Superdex 75 (Cytiva) column, eluted with 1.2 column volumes of the same buffer containing 100 mM NaCl. The Asf1 was repurified by Mono Q chromatography, concentrated, and dialyzed against 20 mM Tris-HCl (pH 7.5), containing 150 mM NaCl, 1 mM dithiothreitol, 0.5 mM EDTA, 0.1 mM PMSF, and 10% glycerol.

## Subcellular fractionation and western blotting

Semi-confluent HeLa cells in a 10 cm dish were chilled on ice and rinsed twice with 5 ml of ice-cold PBF. To permeabilize the cells, 2.5 ml of PBF containing 0.2% Triton X-100 was added, and incubated for 5 min on ice. The supernatant was collected as the soluble fraction. The permeabilized cells were rinsed twice with 5 ml of ice-cold PBF and scraped off the dish. After centrifugation at 800 × g for 5 min at 4℃, the pellets were resuspended in 250 µl of RIPA buffer (50 mM Tris-HCl, pH 7.6, 150 mM NaCl, 1% NP-40, 0.5% sodium deoxycholate, 1 × protease inhibitor cocktail and 0.1% SDS) and treated with benzonase. The supernatant was then collected by centrifugation at 16,000 × g for 5 min at 4℃ as the soluble fraction. The insoluble fraction was diluted ten-fold with PBF containing 0.2% Triton X-100 to adjust the dilution ratio between the soluble and insoluble fractions.

Samples for western blots were separated by SDS-8% PAGE and transferred onto PVDF membranes (Cytiva) with a semi-dry blotter (Bio-Rad) at 120 mA for 60 min. For SRCAP, samples were separated by SDS-6% PAGE and transferred onto PVDF membranes with a wet blotter (Bio-Rad) in an ice-cold 1 × AquaBlot (Wako) solution containing 0.05% SDS, at 120 V for 60 min. After blotting, the membranes were blocked with blocking One (Nacalai), and then probed with the following primary antibodies: anti-HIRA (rabbit, 1/500, Abcam, ab129169), anti-p60 (rabbit, 1/400, Abcam, ab109442), anti-H3 (mouse, 1/1000, MBL, MABI0301), anti-SRCAP (rabbit, 1/1000, Kerafast, ESL103), anti-TIP60 (mouse, 1/50, Santa Cruz, sc-166323), anti-ANP32E (rabbit, 1/1000, MyBioSource, MBS9214243) or anti-GAPDH (mouse, 3/1000, MBL, ML171-3). For the secondary antibodies, sheep horseradish peroxidase (HRP)-conjugated anti-mouse IgG or -rabbit IgG (1/10,000, Cytiva) was used. Signals were developed using Chemi-Lumi One Super (Nacalai) and were detected by an Amersham Imager 680 (Cytiva).

## Acknowledgements

This work was supported by JSPS KAKENHI Grant Numbers JP17H05013, JP16K14785, JP20K06496, JP20H05397, 16H06279 (PAGS) (to HT), JP18H05531, JP18K19310, JP20H03520 (to NS), JP19H04970, JP19H03158, JP20H05393 (to KM), JP18K19432, JP19H05425, JP19H03211, JP20H05368 (to AH), and JP17K19356, JP17H03608, JP18H05527, JP18H04802, JP19H05244, JP20H00456, JP20H04846 (to YO), JP17H01417 and JP18H05527 (to HKimura), JP18H05534, JP20H00449 (to HKurumizaka), JST PRESTO JPMJPR2026 (to KM), JPMJPR19K7 (to AH), AMED JP20ek0109489h0001 (to YO), JST CREST grants JPMJCR16G1 (to YO and HKimura), JST ERATO JPMJER1901, AMED JP20am0101076 (to HKurumizaka). We thank Drs. Yuma Ito at Tokyo Institute of Technology and Kazuhiko Uchida at the Cancer Institute of JFCR for technical advice. We also thank Dr. Crawford at Duke University for the DNaseI-seq data (GEO:GSM816643), Broad Institute for the H2A.Z ChIP-seq data (GEO:GSM1003483) and the ENCODE Consortium. HT is supported by The Nakajima Foundation. NS is supported by the Takeda Science Foundation, the Vehicle Racing Commemorative Foundation and a Research Grant from the Princess Takamatsu Cancer Research Fund.

## Additional information

### Funding

| Funder | Grant reference number | Author |
|---|---|---|
| Japan Society for the Promotion of Science | JP17H05013 | Hiroaki Tachiwana |
| Japan Society for the Promotion of Science | JP16K14785 | Hiroaki Tachiwana |
| Japan Society for the Promotion of Science | JP20K06496 | Hiroaki Tachiwana |
| Japan Society for the Promotion of Science | JP20H05397 | Hiroaki Tachiwana |
| Japan Society for the Promotion of Science | 16H06279 (PAGS) | Hiroaki Tachiwana |
| Japan Society for the Promotion of Science | JP18H05531 | Noriko Saitoh |
| Japan Society for the Promotion of Science | JP18K19310 | Noriko Saitoh |
| Japan Society for the Promotion of Science | JP19H04970 | Kazumitsu Maehara |
| Japan Society for the Promotion of Science | JP19H03158 | Kazumitsu Maehara |
| Japan Society for the Promotion of Science | JP20H05393 | Kazumitsu Maehara |

| | | |
|---|---|---|
| Japan Science and Technology Agency | JPMJPR2026 | Kazumitsu Maehara |
| Japan Society for the Promotion of Science | JP18K19432 | Akihito Harada |
| Japan Society for the Promotion of Science | JP19H05425 | Akihito Harada |
| Japan Society for the Promotion of Science | JP19H03211 | Akihito Harada |
| Japan Society for the Promotion of Science | JP20H05368 | Akihito Harada |
| Japan Science and Technology Agency | JPMJPR19K7 | Akihito Harada |
| Japan Society for the Promotion of Science | JP17K19356 | Yasuyuki Ohkawa |
| Japan Society for the Promotion of Science | JP17H03608 | Yasuyuki Ohkawa |
| Japan Society for the Promotion of Science | JP18H05527 | Yasuyuki Ohkawa Hiroshi Kimura |
| Japan Science and Technology Agency | JMJCR16G1 | Yasuyuki Ohkawa Hiroshi Kimura |
| Japan Society for the Promotion of Science | JP18H04802 | Yasuyuki Ohkawa |
| Japan Society for the Promotion of Science | JP19H05244 | Yasuyuki Ohkawa |
| Japan Society for the Promotion of Science | JP20H00456 | Yasuyuki Ohkawa |
| Japan Society for the Promotion of Science | JP20H04846 | Yasuyuki Ohkawa |
| Japan Agency for Medical Research and Development | JP20ek0109489h0001 | Yasuyuki Ohkawa |
| Japan Society for the Promotion of Science | JP18H05534 | Hitoshi Kurumizaka |
| Japan Science and Technology Agency | JPMJER1901 | Hitoshi Kurumizaka |
| Japan Agency for Medical Research and Development | JP20am0101076 | Hitoshi Kurumizaka |
| Japan Society for the Promotion of Science | JP20H00449 | Hitoshi Kurumizaka |
| Japan Society for the Promotion of Science | JP17H01417 | Hiroshi Kimura |
| Japan Society for the Promotion of Science | JP20H03520 | Noriko Saitoh |
| Nakajima Foundation | | Hiroaki Tachiwana |
| Takeda Science Foundation | | Noriko Saitoh |
| Vehicle Racing Commemorative Foundation | | Noriko Saitoh |
| Princess Takamatsu Cancer Research Fund | | Noriko Saitoh |
| Japan Science and Technology Agency | CREST JPMJCR16G1 | Yasuyuki Ohkawa Hiroshi Kimura |

The funders had no role in study design, data collection and interpretation, or the decision to submit the work for publication.

## Author contributions
Hiroaki Tachiwana, Conceptualization, Data curation, Formal analysis, Supervision, Funding acquisition, Validation, Investigation, Visualization, Methodology, Writing - original draft, Project administration, Writing - review and editing; Mariko Dacher, Yosuke Seto, Ryohei Katayama, Formal analysis, Validation, Investigation, Writing - review and editing; Kazumitsu Maehara, Akihito Harada, Formal analysis, Funding acquisition, Validation, Investigation, Writing - review and editing; Yasuyuki Ohkawa, Funding acquisition, Validation, Investigation, Writing - review and editing; Hiroshi Kimura, Hitoshi Kurumizaka, Conceptualization, Funding acquisition, Validation, Investigation, Writing - review and editing; Noriko Saitoh, Conceptualization, Funding acquisition, Validation, Writing - review and editing

## Author ORCIDs
Hiroaki Tachiwana https://orcid.org/0000-0001-9227-7653
Yasuyuki Ohkawa http://orcid.org/0000-0001-6440-9954
Hiroshi Kimura http://orcid.org/0000-0003-0854-083X

## Decision letter and Author response
Decision letter https://doi.org/10.7554/eLife.66290.sa1
Author response https://doi.org/10.7554/eLife.66290.sa2

# Additional files

## Supplementary files
• Transparent reporting form

## Data availability
The deep sequencing data in this study are available through the NIH GEO Database under the accession numbers GEO: GSE163502, GSE130947 and the DNA DATA bank of Japan under the accession number DDBJ: DRA009580.

The following datasets were generated:

| Author(s) | Year | Dataset title | Dataset URL | Database and Identifier |
|---|---|---|---|---|
| Tachiwana H, Maehara K, Harada A, Ohkawa Y | 2021 | RhIP-ChIP-seq of H2A and H2A.Z using asynchronous, early S and late S phase cells. | https://www.ncbi.nlm.nih.gov/geo/query/acc.cgi?acc=GSE130947 | NCBI Gene Expression Omnibus, GSE130947 |
| Tachiwana H | 2021 | RhIP-ChIP-seq of H2A.Z under ANP32E or ATP depletion condition | https://www.ncbi.nlm.nih.gov/geo/query/acc.cgi?acc=GSE163502 | NCBI Gene Expression Omnibus, GSE163502 |
| Tachiwana H | 2020 | RhIP-ChIP-seq of H3.3 | https://ddbj.nig.ac.jp/DRASearch/submission?acc=DRA009580 | DNA Data Bank of Japan, DRA009580 |

The following previously published datasets were used:

| Author(s) | Year | Dataset title | Dataset URL | Database and Identifier |
|---|---|---|---|---|
| ENCODE DCC | 2011 | Duke_DnaseSeq_HeLa-S3 | https://www.ncbi.nlm.nih.gov/geo/query/acc.cgi?acc=GSM816643 | NCBI Gene Expression Omnibus, GSM816643 |
| ENCODE DCC | 2012 | Broad_ChipSeq_HeLa-S3_H2A.Z | https://www.ncbi.nlm.nih.gov/geo/query/acc.cgi?acc=GSM1003483 | NCBI Gene Expression Omnibus, GSM1003483 |

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
