## [Decision Letter]

**Acceptance summary:**

The method presented in this article, named RhIP, will be of interest for all fields that interface with chromatin dynamics. It provides a new tool to dissect the mechanisms of chromatin assembly and disassembly genome-wide, and to determine how cell cycle and chromatin structure influence these dynamics.

**Decision letter after peer review:**

[Editors’ note: the authors submitted for reconsideration following the decision after peer review. What follows is the decision letter after the first round of review.]

Thank you for submitting your manuscript to *eLife* as a Tools and Resources article. Your article has been reviewed by 3 peer reviewers, one of whom is a member of our Board of Reviewing Editors, and the evaluation has been overseen by a Senior Editor. The reviewers read each other's reviews and discussed their comments.

Although the general enthusiasm is high, significant doubts remain with regard to how faithfully RhIP recapitulates endogenous remodeling activities. In addition, it is unclear how feasible it is to use RhIP to dissect mechanism. For example, can a specific remodeling pathway be blocked by factor depletion? Since substantial amounts of additional work are required to substantiate the conclusions claimed in the paper, we are unable to accept the manuscript in the current form. We are however happy to reconsider if the following concerns are addressed.

1. Given the unique genomic localization of H2A.Z, the authors wisely used H2A.Z as a benchmark to assess the robustness of the RhIP approach. Indeed, site-specific deposition was observed when permeablized cells were incubated with recombinant H2A.Z and (cytosolic) extracts. While this is an interesting result, it remains unclear how well the H2A.Z sites observed by RhIP-ChIP-seq correlate with endogenous H2A.Z sites. The authors should report the percentage of H2A.Z sites correctly identified by RhIP. Scatter plots should be used to compare the relative occupancy of H2A.Z between the RhIP-ChIP-seq data and known H2A.Z sites. Are there H2A.Z sites in the RhIP-ChIP-seq data not seen in vivo. When presenting anecdotal evidence as in Figure 4A, endogenous H2A.Z tracks should be included for comparison.

2. Genomic data analyses generally lack rigor. This is epitomized in Figure 4-supplement 2. The fraction-against-rank plots give little information on the concordance of the two replicates. Do the H2A.Z sites identified in the replicates correlate spatially (e.g. along the genome), quantitatively (e.g. peak height), and qualitatively (e.g. width of peaks)? For example, relative H2A.Z occupancy at known H2A.Z sites of the two H2A.Z RhIP-ChIP-seq replicates should be plotted against each other. The H2A RhIP-ChIP-seq datasets could serve as negative control.

3. It is important to assess how individual factors contribute to a particular remodeling pathway. Given that the steady-state H2A.Z occupancy is a net result of deposition and eviction and that multiple remodeling factors are involved in these processes, whether the observed H2A.Z peaks in RhIP-ChIP-seq represent the consequence of one or more of these factors should be evaluated. For example, are SRCAP and p400/TIP60 responsible for different H2A.Z sites across the genome? Is ATP required for H2A.Z deposition? Is ANP32E responsible for removing H2A.Z at enhancers and insulators? Does transcription contribute to H2A.Z eviction? Proper spike-in controls should be incorporated into the RhIP-ChIP-seq samples before and after the addition of factor-depleted extracts to allow quantitative assessment. Since cytosolic extracts were used, it is not surprising that some factors may be under represented. If such biases do exist, the authors should consider using nuclear extracts and/or provide evidence on how much remodeling factors partitioned into the fractions relative to input (e.g. SRCAP western blots).

4. The title suggests that RhIP is generally applicable to all histones. But there was no mention of the H3 results in the Abstract or the Discussion. In addition, while immunostaining shows H3.1 and H3.3 are incorporated into the nucleus of permeabilized cells at the expected cell cycle stages, microscopy lacks the resolution necessary to validate if H3.1 and H3.3 are targeted to the expected genomic locations. For example, is H3.3 more enriched at active genes? CAF1 and HIRA knockdown should be used to verify if H3.1 and H3.3 are indeed deposited by these pathways. Finally, to show that H3.1 and H3.3 are indeed inserted into nucleosomes, the MNase experiment similar to the one in Figure 3C should be performed.

5. Major improvement of the description of methods is required to allow rigorous assessment of the approach. The current methods section lacks critical details necessary to carry out the experiments.

*Reviewer #1:*

Tachiwana and coworkers have developed a histone deposition assay called RhIP that recapitulates site-specific incorporation of tagged histones into the chromatin of permeabilized cells using cellular extracts. The authors showed that RhIP recapitulates replication-dependent and -independent deposition of histone H3.1 and H3.3, respectively. They also showed that RhIP allows site-specific deposition of H2A, H2A.X and H2A.Z. However, the paper falls short in showing that RhIP can actually be used to dissect molecular mechanisms. For example, if extracts from chaperone/remodeler-depleted cells were used, could a specific histone deposition step be blocked? Could purified proteins then be added back to restore histone deposition activity? In addition, what contribution endogenous factors (i.e. from the permeabilized cells) have on RhIP is unclear.

An *eLife's* Tools and Resources article does not require to report major new biological insights; however, it must be able to demonstrate such advances can be made by applying the new tools. In this respect, whether the RhIP approach can be used to dissect mechanism cannot be evaluated. Therefore, I do not recommend the manuscript in the current form to be published in *eLife*.

Other major issues:

1. 'Open chromatin' and 'close chromatin' are loosely defined terms. The authors should use specific histone marks (e.g. H3K4me3 or H3K27me3) as reference for co-localization analysis of the incorporated histones.

2. Based on the data presented in Figure 4, the robustness of site-specific incorporation of H2A.Z using RhIP cannot be rigorously evaluated. For example, while H2A.Z appears to be deposited at the promoters of some genes, there are H2A.Z peaks outside of promoters. Are these mis-incorporated H2A.Z molecules (i.e. RhIP artifact)? Or are they bona fide H2A.Z sites. The authors should provide endogenous H2A.Z ChIP-seq data for comparison and evaluate how well RhIP recapitulates endogenous H2A.Z deposition.

3. Transfection reagents for proteins, such as the Chariot system, have previously been used to deliver chromatin factors into mammalian cells (Larson AG et al. Nature 2017. PMID:28636604). What is the deposition efficiency of RhIP compared to Chariot?

4. A related question is how efficient is RhIP? What is the percentage of nascent histones incorporated by RhIP? The authors should perform western blots to compare the relative amounts of tagged and untagged histones.

*Reviewer #2:*

The manuscript by Tachinawa et al. presents a new method (called RhIP) that monitors incorporation of histone dimers into permeabilized cells. The authors test H3.1 and H3.3 containing H3-H4 dimers and then they primarily focus on the incorporation of H2A-H2B dimers and their variants containing H2A.X or H2A.Z. The readouts of the assay range from immunofluorescence, to Chromatin IP (ChIP) and ChIP-seq.

Overall, the method is interesting and may become a powerful tool to study chromatin dynamics. However, I have some comments that may improve the current manuscript.

1. As this has been submitted as a Tool and Resource article, I was expecting a much clearer explanation of how the RhIP is carried out. In the method section, every step of the procedure need to be spelled out more clearly. Currently one has to go back to older publications and it becomes quite difficult to recapitulate what has been done in details.

2. If I understand correctly, the authors co-incubate different histone dimers In Figure 1-2 and 5, but not in Figure 3 (and 4 perhaps). This means that the Immunofluorescence results depend on the "competition" between the H2A and the H2A.Z (or h3.1-H3.1) dimers, while the ChIP and ChIP-seq data in figure 3 and 4 only look at incorporation of a single dimer type (without competition for binding between different variants). This makes the comparison of the results quite challenging. What would the result of Figure 1-2 and 5 be if the different histone dimers would be added individually, rather than as a mixture? This would greatly clarify what are the properties that control dimers incorporation in this assay.

3. The authors do not really acknowledge that the incorporation observed in their studies is the net result of assembly AND disassembly pathways. This needs to be part of the narrative, because the experiments do not address which of these pathways is involved and to what extent. For example, on page 11, lines 10-14, or in page 12, line 1-2 or line 18-19, only suppression of deposition mechanisms are taken into account in the interpretation of the data, but a strong disassembly activity here may result in the same effect.

4. The MNase digestions in Figure 3 are very interesting. Can the authors include a higher exposure gel to show that the overall incorporation of Cy5dUTP is the same for H2A and H2A.Z input samples?

5. In Figure 4 C and E (and supplement 1), are the differences between samples statistically significant, or are these only trends? And what about Figure 1D?Please clarify this.

6. The title and the discussion suggest that the RhIP method is generally applicable and validated both for "histones" (as I understand it, it is H3-H4 and H2A-H2B (and variants thereof)). I found the H3-H4 part rather thin compared to the H2A-H2B (and variant) analysis. This is to the point that H3-H4 are completely omitted from the final model of the manuscript. Can the author address this with a title and discussion that focuses on H2A-H2B rather than "histones"?

*Reviewer #3:*

The manuscript "Chromatin structure-dependent histone incorporation revealed by a genome-wide deposition assay" by Tachiwana et al. put-forward a novel method, termed RhIP that allows a detailed investigation of mechanisms fostering histone deposition. In this approach the authors substitute permeabilized cells with recombinantly expressed tagged histones and cytosolic extracts (or chaperone proteins), and then measure histone chromatin deposition using various methods, ranging from immunofluorescence microscopy to RhIP/ChIP-sequencing. While this is an interesting method, some technical questions remain that need to be addressed and clarified. Most importantly, the authors need to show that the factors (chaperones) responsible for a faithful temporal and spatial histone variant deposition (or ejection) are actually still present in the permeabilized cell nucleus or available in the cell extract. The observation that H2A.Z is deposited into open but not condensed chromatin, could be explained by lack of (one) H2A.Z-specific chaperone/remodelling complex(es) after permeabilization. The authors need to demonstrate that these observations are not caused by a rather artificial experimental system that might not resemble the in vivo situation.

My specific concerns are listed in more detail below:

1. There are some clarifications on technical aspects of the RhIP assay missing. As I understood it, this assay was performed with a cytosolic extract, at least this is mentioned in the Materials and methods section. It would be interesting to see whether the known H2A.Z-specific chromatin remodeller complexes, such as SRCAP, p400/TIP60, ANP32E, INO80, etc. that are mostly nuclear proteins and required for H2A.Z deposition/ejection, are also present in this extract. The extracts as well as the cells after permeabilization and reconstitution should be immunoblotted for these chaperones to verify their presence. Additionally, have the authors performed these experiments in the presence or absence of ATP?

2. Figure 2: The authors have shown in Figure 2E that the H2A.Z signal overlaps with the dUTP signal in early but not in late S-phase cells. This result is in contrast to other studies using EdU-staining and microscopy in living cells that observed H2A.Z signals only overlapping with late (constitutive heterochromatin) replicating cells (e.g. Boenisch et al., 2012) and other studies showing H2A.Z levels at transcriptional start sites (euchromatin, early replicating) to be decreased during S-phase (e.g. Nekrasoph et al., 2012, NSMB). This discrepancy needs to be explained and discussed.

3. Figure 5: This is a relevant experiment, and a RhIP-ChIP-seq experiment should be included to show in greater detail that H2A.Z-M6 resembles more H2A than H2A.Z when looking at TSS (see Figure 4B).

[Editors’ note: further revisions were suggested prior to acceptance, as described below.]

Thank you for submitting your article "Chromatin structure-dependent histone incorporation revealed by a genome-wide deposition assay" for consideration by *eLife*. Your article has been reviewed by 3 peer reviewers, one of whom is a member of our Board of Reviewing Editors, and the evaluation has been overseen by Jessica Tyler as the Senior Editor. The reviewers have opted to remain anonymous.

Essential Revisions:

Here we share detailed comments to tackle the limitations of the study:

1. While replication-dependent mechanisms are well captured by RhIP, it is less clear if transcription and chromatin remodeling is functional in this system and thus if transcription-dependent nucleosome exchange processes are faithfully recapitulated. It is important to improve the comparison of RhIP with 'in vivo' (i.e. existing ChIP-seq datasets) localisation and explicitly develop hypotheses why in some cases the data matches the 'in vivo' situation and in others not.

a. Lines 279-301 and Figure 4. There is moderate correlation between genome-wide H2A.Z RhIP-ChIP-seq and H2A.Z ChIP-seq (Figure 4C). However, it seems H2A.Z RhIP-ChIP might be selectively detecting TSS specific signals rather than gene body specific associations (Figure 4A and 4E). It would be helpful to comment on the genetic elements (e.g. gene body) at which H2A.Z RhIP-ChIP and H2A.Z ChIP show low or no correlation and make a comparison. This way strengths and weaknesses of the assay can be better evaluated and compared to already existing protocols. The image resolution of Figure 4A should be improved.

b. Suppl. Figure 1.1 H3.3 distribution across the active and inactive genes were plotted in Suppl Figure 1. However, how active and inactive status of genes was defined, and which data was utilized for this analysis were not indicated under the Methods. These parameters and data source should be included. Transcript levels for GAPDH and LNC02199 should also be indicated or the reader should be referred to a table containing transcription data (i.e. RNA-Seq). I also think that it would be helpful if the authors can comment on the efficiency of H3.3-HA ChIP compared to pull down of endogenous H3.3 by comparing RhIP-ChIP-seq and previously reported H3.3 ChIP-seq analyses.

c. The genomics analysis presented in Figure 3 is somewhat rudimentary, the signals from H2A and H2A.Z appear overall very correlated (Figure 3A, B), albeit the correlation genome-wide calculated in Figure 3-S2 is 'medium'. The scatter plot could be shown better to see potentially different 'population of loci, some that correlate well and 'outliers'. Judging from Figure 3B, it appears that there is a good correlation across broad chromatin states with the notable excavation of active TSS. This is in line with the known H2A.Z mechanism, but could be elaborated in plots a bit clearer. How do the profiles look across highly expressed and non-expressed genes?

The cell-cycle-synchronized plots are only summarized in the heatmap in Figure 3C,D. Example loci could be shown as in Figure 3A.

2. It would be helpful to improve the interpretation of the data to include all existing caveats to the assay setup.

a. The new ATP addition experiments (figure 5E-F) are interpreted as if the only effect of ATP is on chromatin remodellers. It is well possible that by adding ATP in this conditions, additional factors (i.e. transcription, splicing/translation, molecular chaperones, etc) are being affected and responsible for the effect.

b. The authors alternate setups where they either co-incubate different histone dimers, or they use a single histone dimer isoform. It is really unclear to the readers what is the reason for this and how does this affect the results/interpretation. In cases where a mixture of histone dimers were present (i.e. Figures 1A-D, 2, 6), competition between the H2A/H2A.Z-X or H3.1/H3.3 dimers may take place, while in the others this competition is not present. This makes the comparison of the results quite challenging, and their conclusions unclear. This should be discussed at the very least, as it has implications on understanding the properties that control histone incorporation during RhIP.

c. The RhIP-ChIP-seq protocol uses formaldehyde crosslinking instead of native mononucleosomal IP as used in Figure 1H, leaving room for interpretation if the pull-down histone is indeed incorporated into chromatin. Immunofluorescence also does not necessarily prove incorporation of histones into chromatin, since i.e. the fractionation in Figure 1-S2 shows that HIRA and CAF1 remain tightly bound to the isolated nuclei. Thus, accumulation of the epitope-tagged H3 in the nucleus of cells could merely mean that is bound to the histone chaperones present in the nucleus (competing away endogenous histones). It is not possible to discern from the data what the incorporated amount versus the chaperone-bound fraction of recombinant histone is.

d. The functional assessment of ANP32E knockdown lacks statistical analysis – clearly the quite robust knockdown (Figure 5B) does not majorly abrogate TSS incorporation of H2A.Z. Is the difference statistically significant? What is the explanation for the relatively small effect size?

e. Lines 275-277. Please revisit the conclusion and make the statement clearer on H2A.Z incorporation and elimination.

We suggest adding these references:

The authors make a thorough effort to place their method in the context of existing pulse-labeling methods, but Tet-ON based pulsing methods, which do give similar results have not been discussed: Mito, Y., Henikoff, J. G. and Henikoff, S. Genome-scale profiling of histone H3.3 replacement patterns. Nat. Genet. 37, 1090-1097 (2005); Yildirim, O. et al. A system for genome-wide histone variant dynamics in ES cells reveals dynamic MacroH2A2 replacement at promoters. PLoS Genet. 10, e1004515 (2014); Ha, M., Kraushaar, D. C. and Zhao, K. Genome-wide analysis of H3.3 dissociation reveals high nucleosome turnover at distal regulatory regions of embryonic stem cells. Epigenetics Chromatin 7, 38 (2014).

Additional notes (not needed for revision):

Finally we list additional suggestions that were made by the reviewers. These comments are shared with you, but the reviewers do not expect these experiments to be performed in the revised version of the manuscript. We hope these will help you improve the discussion part and future studies.

Figure 1

Deposition of H3.3 seems rather inefficient as judged from the genome-wide profiling RhIP-ChIP-seq experiment in Figure 1-S1, and deposition appears to only occur at TSS but not across the gene bodies of active genes as observed in cells (please make sure to provide primary data on GEO for a revised submission). As I understand, the RhIP-ChIP-seq protocol uses formaldehyde crosslinking instead of native mononucleosomal IP as used in Figure 1H, leaving room for interpretation if the pull-down histone is indeed incorporated into chromatin. Immunofluorescence also does not necessarily prove incorporation of H3.1/H3.3 into chromatin, since the fractionation in Figure 1-S2 shows that HIRA and CAF1 remain tightly bound to the isolated nuclei. Thus, accumulation of the epitope-tagged H3 in the nucleus of cells could merely mean that is bound to the histone chaperones present in the nucleus (competing away endogenous histones). It is not possible to discern from the data what the incorporated amount versus the chaperone-bound fraction of recombinant histone is. Potentially, the product of the RhIP reaction could itself be fractionated using e.g. a salt gradient, which would provide evidence if the retained H3 histones are nucleosomal (retained up to ~500mM NaCl) or non-nucleosomal and dissociating at lower salt concentrations.

Further, in Figure 1H shoes that Cy5-labeled mononucleosomes are at very low abundance within the input (almost undetectable Cy5 signal in lanes 2,3,4) but highly enriched in both H3.1 and H3.3 IP. The fraction of replicated chromatin in each lane can be estimated by the Cy5/SYBR gold ratio. Assuming completely random (i.e. replication-independent) incorporation of recombinant histones, the Cy5/SYBR gold ration after IP (lanes 6,7) should be equivalent to that ratio before IP (lanes 4,5,6). Yet, in lanes 6,7, there's much more Cy5 signal thus, a strong enrichment for replicating chromatin. This suggest that not only H3.1 but also a considerable fraction of H3.3 is incorporated replication-dependent in this assay. This is entirely possible given the known role of HIRA/H3.3 in gap-filling after replication fork, especially if normal replication-dependent pathways are impaired (Ray-Gallet et al., Mol Cell, 2011). Again it needs to be stressed that the IF experiment Figure 1C do not necessarily prove that H3.3 is incorporated into nucleosomes in cells not undergoing replication.

Lastly, the knockdown experiments in Figure 1-S2 appear to confirm known roles of CAF-1 and HIRA, but important controls have and opportunities to both validate the system and learn something new have been missed: following the flowchart of Figure 1-S2B, it is evident that there's ample opportunity to test the authors hypothesis that RhIP assay indeed recapitulates the assembly pathways known in cells. For example, CAF-1 and HIRA knockdown should each be tested for epitope tagged H3.1 and H3.3, to exclude the possibility that what we are looking at here is chromatin assembly and not mere binding to nuclear chaperones. Furthermore, replication can only proceed in the presence of all four deoxynucleotides whereas chromatin remodeling requires only ATP. Thus by adding, or not, nucleotides, it can be explicitly tested if active replication is needed. Consider following setup and expected results:

no siRNA + dNTPs + ATP → nuclei are positive for H3.1 and H3.3 (shown in Figure 1-S2)

no siRNA – dNTPs + ATP → no replication! So nuclei should be negative for H3.1 unless the H3.1 signal is actually coming from H3.1 binding to CAF1. But H3.3 incorporation should be unaffected if truly replication-independent!

no siRNA – dNTPs – ATP → in the absence of nucleotides and ATP there is no energy to remove existing nucleosomes. Since we have to assume that DNA in isolated nuclei is normally chromatinized, chromatin assembly with epitope-tagged histones H3 would require prior nucleome eviction. This cannot happened without energy. Thus, in this control nuclei should be negative for H3.3. If they are still positive, then H3.3 is most likely bound by nuclear chaperones but not incorporated.

siCAF + dNTPs + ATP → nuclei are negative for H3.1 (shown in Figure 1-S2) but should be positive for H3.3 (not assayed). If gap-filling assembly via HIRA is active in RhIP assay, than H3.3 signal should even increase upon CAF1 depletion!

siHIRA + dNTPs + ATP → nuclei are negative for H3.3 (shown in Figure 1-S2) but should be remain for H3.1 when in replication (not assayed).

Figure 1 figure supplement 1B: It would make the argument stronger if the authors could show that this difference is not observed for H3.1-containing dimers.

Figure 1 figure supplement 1B: It would make the argument stronger if the authors could show that this difference is not observed for H3.1-containing dimers.

Figure 2

Since there is no negative control present for H2A, H2A.X, H2A.Z incorporation (all nuclei are positive irrespective of cell cycle), there remains an more pertinent concern that some or most of the fluorescent signal observed after RhIP could arise from non-incorporated histones that accumulate with chaperones in the nucleus. Again, a biochemical fractionation could cleanly show this, potentially also a pre-extraction protocol before immunostaining. A negative control (H2A or H2A.Z chaperone knockdown that abrogates fluorescent signal) should be established, or a -ATP control as used for the genomics experiment in Figure 5.

In Figure 2F, as with Figure 1H, it is evident that replicated chromatin is overrepresented in both H2A and H2A.Z pulldowns, suggesting that both are deposited also in a replication-dependent manner (also supported by Figure 2D image quantification). Comparison to the very similar looking fold-enrichment from pure synchronized S phase population Figure 2-S2 actually may suggest that most incorporation into proper nucleosomes is replication-dependent in this assay.

Figure 5

Is ANP32E acting catalytically (i.e. small amounts would be sufficient for keeping up eviction activity)? What's maybe more surprising is that ATP depletion is not really effective. Histone eviction requires ATP. Thus this mean there are many pre-existing nucleosome-free regions? Or could this suggest that much of the signal observed does not correspond to properly assembled nucleosomes?

Given the hypothesis put forward from Figure 5 (ATP-dependent remodeling and ANP32E together facilitate H2A.Z incorporation), then it should be tested if combined ANP32E + ATP depletion has an synergistic, additive or no additional effect beyond the single treatments.

Figure 6

The image-based analysis does support the conclusion that the M6 mutant behaves H2A-like, but it is unclear what the mechanistic insight gained from this experiment is, again acknowledging that this mutant has been constructed according to known reports.

Figure 1H: beyond demonstrating that the epitope-tagged histone associates with nucleosomal-sized DNA, a protein gel or blot would be revealing to demonstrate a stoichiometric relationship of H3, H4, H2A, H2B. A very interesting question could be addressed by assessing the ratio of epitope-tagged H3.1/3 to untagged H3 in these nucleosomal fragments: do replication-dependent and independent deposition pathways assemble homotypic H3/H4 tetramers with only exogenous (tagged) H3.1, H3.3? Or is the tagged histone H3 mixed equally with endogenous (untagged) histone H3, thus suggesting that one new H3/H4 dimer is paired with an existing H3/H4 dimer during the assembly process.

I would highly recommend performing time-course experiments to really understand the kinetics, dynamic range and saturation of the system.

---

## [Author Response]

[Editors’ note: the authors resubmitted a revised version of the paper for consideration. What follows is the authors’ response to the first round of review.]

Although the general enthusiasm is high, significant doubts remain with regard to how faithfully RhIP recapitulates endogenous remodeling activities. In addition, it is unclear how feasible it is to use RhIP to dissect mechanism. For example, can a specific remodeling pathway be blocked by factor depletion? Since substantial amounts of additional work are required to substantiate the conclusions claimed in the paper, we are unable to accept the manuscript in the current form. We are however happy to reconsider if the following concerns are addressed.1) Given the unique genomic localization of H2A.Z, the authors wisely used H2A.Z as a benchmark to assess the robustness of the RhIP approach. Indeed, site-specific deposition was observed when permeablized cells were incubated with recombinant H2A.Z and (cytosolic) extracts. While this is an interesting result, it remains unclear how well the H2A.Z sites observed by RhIP-ChIP-seq correlate with endogenous H2A.Z sites.

Thank you for this comment. We agree that it is important to clarify how well the H2A.Z sites observed by RhIP-ChIP-seq correlate with the endogenous H2A.Z sites detected by ChIP-seq. Accordingly, we have reanalyzed our data, and made new figures in our revised manuscript. To clearly answer this question, we separated your comments and answered each one, as below.

The authors should report the percentage of H2A.Z sites correctly identified by RhIP. Are there H2A.Z sites in the RhIP-ChIP-seq data not seen in vivo.

According to this comment, we visualized the H2A.Z sites of the RhIPChIP-seq read density across the endogenous H2A.Z peaks determined by ChIP-seq, using heatmaps (new Figure 4D and Figure 4—figure supplement 1A). We found that the H2A.Z incorporated by the RhIP assay exists in almost all endogenous H2A.Z peaks, while the H2A is excluded from the H2A.Z peaks. The reversed heatmap, which depicts the H2A.Z of the ChIP-seq read density across the H2A.Z peaks determined by the RhIP-ChIP-seq, showed that only 3% of the H2A.Z peaks are specific for the RhIP assay (new Figure 4E and Figure 4—figure supplement 1B). We described these analyses on page 15, lines 297-303.

Scatter plots should be used to compare the relative occupancy of H2A.Z between the RhIP-ChIP-seq data and known H2A.Z sites.

We drew scatter plots showing the comparison between the RhIP-ChIP-seq and ChIP-seq data at known H2A.Z sites. We found a moderately linear relationship between H2A.Z (RhIP-ChIP-seq) and H2A.Z (ChIP-seq), but little to no correlation between H2A (RhIP-ChIP-seq) and (ChIP-seq) (new Figure 4C). We described this on page 15, lines 293-297.

2) Genomic data analyses generally lack rigor. This is epitomized in Figure 4-supplement 2. The fraction-against-rank plots give little information on the concordance of the two replicates. Do the H2A.Z sites identified in the replicates correlate spatially (e.g. along the genome), quantitatively (e.g. peak height), and qualitatively (e.g. width of peaks)? For example, relative H2A.Z occupancy at known H2A.Z sites of the two H2A.Z RhIP-ChIP-seq replicates should be plotted against each other. The H2A RhIP-ChIP-seq datasets could serve as negative control.

Thank you for this suggestion. In the revised manuscript, we confirmed the reproducibility of the RhIP-ChIP-seq of H2A.Z according to this suggestion. We plotted the H2A.Z RhIP-ChIP-seq replicates at known H2A.Z sites, which showed a quite high correlation (0.95 Pearson correlation coefficient), while the H2A and H2A.Z RhIP-ChIP-seq data showed a weaker correlation, as compared to one of the replicates (new Figure 3—figure supplement 2A). Moreover, the aggregation plots of the two H2A.Z RhIP-ChIP-seq replicates at known H2A.Z sites showed similar accumulation patterns (new Figure 4D and new Figure 4—figure supplement 1A). In contrast, the aggregation plot of the H2A RhIP-ChIPseq showed the exclusion of H2A at the known H2A.Z sites (new Figure 4D). We described these findings on page 18, lines 377-382.

3) It is important to assess how individual factors contribute to a particular remodeling pathway. Given that the steady-state H2A.Z occupancy is a net result of deposition and eviction and that multiple remodeling factors are involved in these processes, whether the observed H2A.Z peaks in RhIP-ChIP-seq represent the consequence of one or more of these factors should be evaluated. For example, are SRCAP and p400/TIP60 responsible for different H2A.Z sites across the genome? Is ATP required for H2A.Z deposition?

According to these suggestions, we first examined the permeabilized cells for the presence of SRCAP, TIP60 and ANP32E. The fractionation and following western blot analysis confirmed that SRCAP and TIP60 are still in the permeabilized cells, suggesting that the chromatin remodeling activity may be involved in the H2A.Z deposition in the RhIP assay. Since chromatin remodeling requires ATP, we then prepared the ATP-reduced cellular extract and tested it in the RhIP assay with or without ATP supplementation (new Figures 5E and F). As a result, ATP promoted the H2A.Z depositions in active TSS and enhancer regions, suggesting that the H2A.Z deposition in the RhIP assay requires a chromatin remodeling activity that hydrolyzes ATP. We described these new results on pages 1617, lines 329-341.

Is ANP32E responsible for removing H2A.Z at enhancers and insulators? Does transcription contribute to H2A.Z eviction?

Our fractionation and western blot analysis showed that ANP32E is removed from permeabilized cells during the permeabilization process (new Figure 5A). As ANP32E is present in the cellular extract, which is used in the RhIP assay, we prepared the ANP32E-knockdown cellular extract and performed a RhIP-ChIP-seq analysis of H2A.Z (new Figures 5B-D). As a result, the efficiency of H2A.Z incorporations into active TSS and enhancer regions, where endogenous H2A.Z is predominantly localized, was decreased upon the ANP32E-knockdown (new Figure 5D and Figure 5—figure supplement 1A). This indicates that the ANP32Edependent evictions of the pre-incorporated H2A.Z in active TSS and enhancer regions promotes the incorporation of exogenously added H2A.Z in the RhIP assay. We described these data on page 16, lines 313-328.

We note that a previous paper reported that ANP32E-knockout mouse embryonic fibroblast (MEF) cells showed an increased number of H2A.Z peaks in insulator regions. In the RhIP assay using the ANP32Eknockdown cellular extract, the changes in the efficiency of the H2A.Z deposition in the insulator regions were less than those in the active TSS and enhancers (new Figure 5D). This discrepancy may arise from the fact that the H2A.Z distribution pattern of the knockout MEFs represents chromatin reorganization over a long time during embryogenesis, thus facilitating the adaptation to the ANP32E-knockout condition. This may include compensation for the ANP32E-mediated H2A.Z evictions. In contrast, our RhIP-seq analysis using the ANP32E-knockdown cellular extract instead represents the chromatin dynamics after the acute depletion of ANP32E. Consistent with our data, another previous paper showed that the introduction of recombinant ANP32E to zebrafish embryos resulted in the global loss of H2A.Z from the nucleus (P. J. Murphy, Wu, James, Wike, and Cairns, 2018).

Murphy, P. J., Wu, S. F., James, C. R., Wike, C. L., and Cairns, B. R.

(2018). Placeholder Nucleosomes Underlie Germline-to-Embryo DNA

Methylation Reprogramming. *Cell*, *172*(5), 993–1006.e13. http://doi.org/10.1016/j.cell.2018.01.022

Proper spike-in controls should be incorporated into the RhIP-ChIP-seq samples before and after the addition of factor-depleted extracts to allow quantitative assessment.

Thank you for this critical comment. Accordingly, we performed the ATPand ANP32E-depleted RhIP-ChIP-seq analysis using the spike-in controls (new Figures 5C-F and Figure 5—figure supplement 1) (Chen et al., 2015; Egan et al., 2016; Orlando et al., 2014).

Chen, K., Hu, Z., Xia, Z., Zhao, D., Li, W., and Tyler, J. K. (2015). TheOverlooked Fact: Fundamental Need for Spike-In Control for Virtually All Genome-Wide Analyses. Molecular and Cellular Biology, 36(5), 662–667. http://doi.org/10.1128/MCB.00970-14

Egan, B., Yuan, C.-C., Craske, M. L., Labhart, P., Guler, G. D., Arnott, D., et al. (2016). An Alternative Approach to ChIP-seq Normalization Enables Detection of Genome-Wide Changes in Histone H3 Lysine 27 Trimethylation upon EZH2 Inhibition. PLoS ONE, 11(11), e0166438. http://doi.org/10.1371/journal.pone.0166438

Orlando, D. A., Chen, M. W., Brown, V. E., Solanki, S., Choi, Y. J., Olson, E. R., et al. (2014). Quantitative ChIP-seq normalization reveals global modulation of the epigenome. Cell Reports, 9(3), 1163–1170. http://doi.org/10.1016/j.celrep.2014.10.018

Since cytosolic extracts were used, it is not surprising that some factors may be under represented. If such biases do exist, the authors should consider using nuclear extracts and/or provide evidence on how much remodeling factors partitioned into the fractions relative to input (e.g. SRCAP western blots).

As mentioned above, our fractionation analysis indicated that SRCAP and TIP60, components of major H2A.Z deposition complexes, were retained in the permeabilized cells. Even though ANP32E was extracted during the permeabilization process, ANP32E was added back to the RhIP reaction as the ANP32E present in the cellular extract. These data indicate that these factors are not underrepresented in the RhIP assay.

Together with the results mentioned above, we conclude that the H2A.Z deposition in the RhIP assay is a net result of the ANP32E-mediated eviction and deposition of H2A.Z and the replacement of pre-incorporated H2A with H2A.Z by chromatin remodeling, and that the cellular H2A.Z deposition pathways are recapitulated in the RhIP assay.

4) The title suggests that RhIP is generally applicable to all histones. But there was no mention of the H3 results in the Abstract or the Discussion.

Thank you for this advice. We added sentences regarding the H3 results in the Abstract and Discussion on page 2, lines 6-8 and page 19, lines 390-396.

In addition, while immunostaining shows H3.1 and H3.3 are incorporated into the nucleus of permeabilized cells at the expected cell cycle stages, microscopy lacks the resolution necessary to validate if H3.1 and H3.3 are targeted to the expected genomic locations. For example, is H3.3 more enriched at active genes?

We agree with this comment. We performed the RhIP-ChIP-seq analysis of H3.3 and found its enrichment at active genes (new Figure 1—figure supplement 1A). In addition, the RhIP-ChIP-qPCR analysis showed higher enrichment of H3.3 in the active gene (GAPDH) than the inactive gene (LINC02199) (new Figure 1—figure supplement 1B). We described these data on pages 8 and 9, lines 143-162.

CAF1 and HIRA knockdown should be used to verify if H3.1 and H3.3 are indeed deposited by these pathways.

We confirmed the CAF-1 and HIRA requirements for the H3.1 and H3.1 depositions in the RhIP assay by immunostaining (new Figure 1—figure supplement 2), since previous SNAP-tag based analyses clearly showed their requirement by imaging (Ray-Gallet et al., 2011). First, we checked for the existence of CAF-1 and HIRA in permeabilized cells. Our fractionation followed by western blotting analysis showed that these factors were retained in permeabilized cells (new Figure 1—figure supplement 2A). We then prepared the CAF-1- or HIRA-knockdown permeabilized cells and performed the RhIP-immunostaining assay of H3.1 or H3.3 (new Figure 1—figure supplements 2B-D). In this assay, we co-cultured the control and knockdown cells on the same coverslips for a precise evaluation. The knockdown cells were judged by immunostainings for CAF-1 (p60) and HIRA. As a result, we found that the intensities of both the incorporated H3.1 and H3.3 were significantly decreased in the CAF-1- and HIRA-knockdown cells, respectively. Therefore, H3.1 and H3.3 are deposited by the CAF-1- and HIRA-dependent pathways in the RhIP assay, respectively. We described these results on pages 9 and 10, lines 163-180.

Ray-Gallet, D., Woolfe, A., Vassias, I., Pellentz, C., Lacoste, N., Puri, A., et al. (2011). Dynamics of histone H3 deposition in vivo reveal a nucleosome gap-filling mechanism for H3.3 to maintain chromatin integrity. Molecular Cell, 44(6), 928–941.

Finally, to show that H3.1 and H3.3 are indeed inserted into nucleosomes, the MNase experiment similar to the one in Figure 3C should be performed.

According to this comment, we performed an MNase experiment similar to that in the previous Figure 3C (currently Figure 2F), using H3.1 and H3.3 (new Figures 1F-H). As a result, we confirmed their incorporations into the chromatin of permeabilized cells. Moreover, we found that the amount of nascent DNA in the H3.1 RhIP-ChIP sample was more than that in the H3.3 RhIP-ChIP sample. This is consistent with the results of the RhIP immunostaining assay shown in the new Figures 1C-E. We described these data on pages 8 and 9, lines 143-154.

5) Major improvement of the description of methods is required to allow rigorous assessment of the approach. The current methods section lacks critical details necessary to carry out the experiments.

We rewrote the methods section on page 23, lines 480-482, pages 24-30, lines 495-637, page 31, lines 658-672 and page 32, lines 701-724 in the revised manuscript. We hope the new methods section is sufficient for readers to perform the RhIP assay. If more information is needed, we will be happy to provide any details. We then express this in Data availability as If more information, or tips are needed, we will/are willing to provide upon reasonable requests.

[Editors’ note: what follows is the authors’ response to the second round of review.]

Essential Revisions:Here we share detailed comments to tackle the limitations of the study:1. While replication-dependent mechanisms are well captured by RhIP, it is less clear if transcription and chromatin remodeling is functional in this system and thus if transcription-dependent nucleosome exchange processes are faithfully recapitulated. It is important to improve the comparison of RhIP with 'in vivo' (i.e. existing ChIP-seq datasets) localisation and explicitly develop hypotheses why in some cases the data matches the 'in vivo' situation and in others not.

We thank the reviewers for this important comment. Based on the suggestions, we performed additional analyses as described below, which allowed us to hypothesize that the difference between the RhIP-ChIP-seq and ChIP-seq may reflect the recycling frequency of the evicted histone complexes.

a. Lines 279-301 and Figure 4. There is moderate correlation between genome-wide H2A.Z RhIP-ChIP-seq and H2A.Z ChIP-seq (Figure 4C). However, it seems H2A.Z RhIP-ChIP might be selectively detecting TSS specific signals rather than gene body specific associations (Figure 4A and 4E). It would be helpful to comment on the genetic elements (e.g. gene body) at which H2A.Z RhIP-ChIP and H2A.Z ChIP show low or no correlation and make a comparison. This way strengths and weaknesses of the assay can be better evaluated and compared to already existing protocols.

According to this suggestion, we have re-analyzed our data and created a new scatter plot between the RhIP-ChIP-seq and ChIP-seq data of H2A.Z at known H2A.Z sites in gene bodies, exclusively (new Figure 4C right). As this reviewer pointed out, we found that there is a fairly linear relationship between them in the gene bodies, indicating that the RhIP-ChIP-seq preferentially detects TSS-specific incorporation. This may have originated from the fact that H2A.Z can be exchanged more dynamically at TSS, as compared to gene bodies. Alternatively, as the histones that are evicted during transcription can be recycled (1, 2), the lower correlation between RhIP-ChIP and ChIP of H2A.Z in gene bodies may reflect the lower frequency of de novo H2A.Z deposition and the higher frequency of H2A.Z recycling during transcription. We described these analyses on page 15, lines 301-304 and discussed them on page 22, lines 453-457.

1. Torné J, Ray-Gallet D, Boyarchuk E, Garnier M, Le Baccon P, Coulon A, Orsi GA, Almouzni G. (2020) Two HIRA-dependent pathways mediate H3.3 de novo deposition and recycling during transcription. Nature Structural Molecular Biology 27(11):1057-1068. doi: 10.1038/s41594-020-0492-7.

2. Jeronimo, C., Poitras, C., and Robert, F. (2019). Histone Recycling by FACT and Spt6 during Transcription Prevents the Scrambling of Histone Modifications.

Cell Reports, 28(5), 1206–1218.e8. http://doi.org/10.1016/j.celrep.2019.06.097

The image resolution of Figure 4A should be improved.

We improved the image resolution of Figure 4A, from 144 dpi (previous) to 300 dpi (current). Please note that all of the bigwig files shown in this study were created with 1 bp intervals on the genome and not smoothed. This may be the reason why the image appears to have low resolution as compared with the smoothed bigwig files.

b. Suppl. Figure 1.1 H3.3 distribution across the active and inactive genes were plotted in Suppl Figure 1. However, how active and inactive status of genes was defined, and which data was utilized for this analysis were not indicated under the Methods. These parameters and data source should be included.

We thank the reviewer for this comment. We defined genes with 5 or more read counts in the RNA-seq data (GSE140768) as expressed genes and the rest as non-expressed genes. We now describe these parameters in the Methods on page 33, lines 698-700.

Transcript levels for GAPDH and LNC02199 should also be indicated or the reader should be referred to a table containing transcription data (i.e. RNA-Seq).

We have created the suggested table, and now show it in the new Figure 1—figure supplement 1C.

I also think that it would be helpful if the authors can comment on the efficiency of H3.3-HA ChIP compared to pull down of endogenous H3.3 by comparing RhIP-ChIP-seq and previously reported H3.3 ChIP-seq analyses.

According to this suggestion, we assessed the signal-to-noise ratios as part of the efficiency of the methods. We have calculated and compared the S/N ratios of RhIP-ChIP-seq and the previously reported ChIP-seq of H3.3, and created new figures (new Figure 1—figure supplements 1D and E). We found that the S/N ratios are almost the same. We now describe these results on page 11, lines 199-201.

c. The genomics analysis presented in Figure 3 is somewhat rudimentary, the signals from H2A and H2A.Z appear overall very correlated (Figure 3A, B), albeit the correlation genome-wide calculated in Figure 3-S2 is 'medium'. The scatter plot could be shown better to see potentially different 'population of loci, some that correlate well and 'outliers'.

We thank the reviewer for this important comment. First, we would like to explain that the analysis presented in Figure 3A shows the entire genome-wide pattern. On the other hand, the scatter plots in Figure 3—figure supplement 2A were calculated at H2A.Z peaks detected in ChIP-seq. As this reviewer pointed out, H2A.Z and H2A are differently incorporated at H2A.Z peaks locally, as the correlation is moderate (Pearson correlation is 0.48). This is consistent with the results shown as the heatmap in Figure 4D, in which the H2A of RhIP-ChIP-seq showed no accumulation at the H2A.Z peaks of ChIP-seq. On the other hand, they are globally incorporated with certain similarity (Pearson correlation is 0.62, new Figure 3A right). We describe this on page 13, line 257 and page 16, lines 307-309.

Judging from Figure 3B, it appears that there is a good correlation across broad chromatin states with the notable excavation of active TSS. This is in line with the known H2A.Z mechanism, but could be elaborated in plots a bit clearer. How do the profiles look across highly expressed and non-expressed genes?

Thank you for this suggestion. We have recalculated our data and revised the aggregation plot to show the expressed and non-expressed genes separately (new Figure 4B). It is now clear that H2A.Z predominantly accumulated at the TSS of active (expressed) genes. We described this on page 15, line 296.

The cell-cycle-synchronized plots are only summarized in the heatmap in Figure 3C,D. Example loci could be shown as in Figure 3A.

In the revised manuscript, we show the example loci in the new Figure 3—figure supplement 1B.

2. It would be helpful to improve the interpretation of the data to include all existing caveats to the assay setup.

Thank you for this suggestion. We addressed this comment, as described below.

a. The new ATP addition experiments (figure 5E-F) are interpreted as if the only effect of ATP is on chromatin remodellers. It is well possible that by adding ATP in this conditions, additional factors (i.e. transcription, splicing/translation, molecular chaperones, etc) are being affected and responsible for the effect.

We agree with this comment. We discussed the possible effects of ATP addition, on page 18 lines 353-355.

b. The authors alternate setups where they either co-incubate different histone dimers, or they use a single histone dimer isoform. It is really unclear to the readers what is the reason for this and how does this affect the results/interpretation. In cases where a mixture of histone dimers were present (i.e. Figures 1A-D, 2, 6), competition between the H2A/H2A.Z-X or H3.1/H3.3 dimers may take place, while in the others this competition is not present. This makes the comparison of the results quite challenging, and their conclusions unclear. This should be discussed at the very least, as it has implications on understanding the properties that control histone incorporation during RhIP.

We agree with this reviewer’s concern. We understand that the proper combination of histone chaperones and histone variants ensures the correct localizations of histone variants. However, if one histone variant is increased, then the excess histone variant may bind to unsuitable histone chaperones, leading to ectopic localization. Indeed, the overexpressed histone H3 variant, CENP-A, in cells aberrantly binds to the H3.3 chaperone, DAXX, leading to ectopic localizations (1). In this study, we performed the RhIP assay with two different histone variants simultaneously, which do not share the same chaperone. Therefore, if the amounts of added histone complexes are appropriate, then they would only interact with their defined chaperones and be deposited in a specific manner. As a result, we observed different staining patterns in the same nucleus when H3.1 and H3.3 or H2A/H2A.X and H2A.Z were co-incubated (Figures 1A-D, 2, 6), which proved that they were incorporated into the chromatin by specific mechanisms, rather than non-specifically incorporated. These results also indicate that there is little to no competition under these conditions. Therefore, the results are unlikely to change if one or two histone variant(s) was/were used for the RhIP assay. However, as the reviewer mentioned, the competition assay using this system is worth trying, to analyze the overexpression situations of a specific histone variant. We added this discussion on page 20, lines 402-412.

1. Lacoste N, Woolfe A, Tachiwana H, Garea AV, Barth T, Cantaloube S, Kurumizaka H, Imhof A, Almouzni G (2014) Mislocalization of the centromeric histone variant CenH3/CENP-A in human cells depends on the chaperone DAXX Molecular Cell 53(4):631-44. doi: 10.1016/j.molcel.2014.01.018.

c. The RhIP-ChIP-seq protocol uses formaldehyde crosslinking instead of native mononucleosomal IP as used in Figure 1H, leaving room for interpretation if the pull-down histone is indeed incorporated into chromatin. Immunofluorescence also does not necessarily prove incorporation of histones into chromatin, since i.e. the fractionation in Figure 1-S2 shows that HIRA and CAF1 remain tightly bound to the isolated nuclei. Thus, accumulation of the epitope-tagged H3 in the nucleus of cells could merely mean that is bound to the histone chaperones present in the nucleus (competing away endogenous histones). It is not possible to discern from the data what the incorporated amount versus the chaperone-bound fraction of recombinant histone is.

Thank you for this comment. This point is critical and must be clarified experimentally. We therefore performed an additional RhIP-immunostaining assay. We treated the cells with PBST containing 300 mM NaCl after the RhIP reaction, to wash out the nuclear proteins that are associated with chromatin, and then fixed the cells. In fact, CAF-1 and HIRA were no longer detected by the immunostaining; however, the exogenously added H3.1 and H3.3 were still detected in the permeabilized cells. This result indicates that the exogenously added H3.1 and H3.3 were mostly incorporated into the chromatin of the permeabilized cells in the RhIP assay, as shown by the native ChIP experiment. We added these data in the new Figure 1—figure supplement 3 and described them on page 10, lines 175-184.

d. The functional assessment of ANP32E knockdown lacks statistical analysis – clearly the quite robust knockdown (Figure 5B) does not majorly abrogate TSS incorporation of H2A.Z. Is the difference statistically significant?

We thank the reviewer for pointing out this issue. We think the TSS incorporation of H2A.Z. significantly changed upon the ANP32E knockdown, with a p-value below 10^-50^. We would like to be cautious about mentioning this, because we included more than forty thousand loci for each in our analysis. With such a large number of samples, the p-value tends to be close to zero; therefore, it does not necessarily show practical significance (1).

1. Lin, M., Lucas, H. C., Jr., and Shmueli, G. (2013). Too big to fail: Large samples and the p-value problem. Information Systems Research, 24(4), 906–917. https://doi.org/10.1287/isre.2013.0480

What is the explanation for the relatively small effect size?

We agree with the reviewer that the reduction of TSS incorporation of H2A.Z is relatively small in the ANP32E knockdown cellular extract. We suggest the presence of a redundant mechanism, which is consistent with the previous report that ANP32E knockout mice did not exhibit any apparent effects on their viability (1), even though proper H2A.Z incorporation is essential for cell survival and proliferation. We also found that ATP promotes the H2A.Z depositions in the RhIP assay. Since ATP is essential for many biological processes, such as ATP-dependent chromatin remodeling and transcription, the H2A.Z deposition in the RhIP assay may require ATP-dependent processes. Taken together, the H2A.Z deposition in the RhIP assay is the net result of the ANP32E-mediated eviction and deposition of H2A.Z, and the replacement of pre-incorporated H2A with H2A.Z by ATP-dependent processes. This is our explanation for the limited effects of the ANP32E knockdown on the H2A.Z incorporation. We discussed this on pages 21-22, lines 443-457.

1. Reilly PT, Afzal S, Wakeham A, Haight J, You-Ten A, Zaugg K, Dembowy J, Young A, Mak TW. (2010) Generation and characterization of the Anp32e-deficient mouse PLoS One, 5(10):e13597. doi: 10.1371/journal.pone.0013597.

e. Lines 275-277. Please revisit the conclusion and make the statement clearer on H2A.Z incorporation and elimination.

According to this comment, we rewrote the conclusion, on page 15, lines 284-286.

We suggest adding these references:The authors make a thorough effort to place their method in the context of existing pulse-labeling methods, but Tet-ON based pulsing methods, which do give similar results have not been discussed: Mito, Y., Henikoff, J. G. and Henikoff, S. Genome-scale profiling of histone H3.3 replacement patterns. Nat. Genet. 37, 1090-1097 (2005); Yildirim, O. et al. A system for genome-wide histone variant dynamics in ES cells reveals dynamic MacroH2A2 replacement at promoters. PLoS Genet. 10, e1004515 (2014); Ha, M., Kraushaar, D. C. and Zhao, K. Genome-wide analysis of H3.3 dissociation reveals high nucleosome turnover at distal regulatory regions of embryonic stem cells. Epigenetics Chromatin 7, 38 (2014).

Thank you for this suggestion. We have already cited the first paper on page 9 line 155, to refer to H3.3 incorporation in the genome. We newly added the second and third references to introduce the tetracycline-based methods, on page 6, lines 99-101.